# Single-cell exon deletion profiling reveals splicing events that shape gene expression and cell state dynamics

Bandana Kumari[1,6], Arun Prasath Damodaran[1,6], Wilfried M. Guiblet[1,2,6], Mei-Sheng Xiao[1], Amit K. Behera [1], Tyler A. On[3], Carl E. McIntosh [1,4], Maxwell Teszler[1], Chelsee Holloway[3], Sandra Le[1], Nikhil Parab[1], Yongmei Zhao [5], Michael Aregger [3] ✉ & Thomas Gonatopoulos-Pournatzis [1] ✉

Alternative splicing is a pervasive gene regulatory mechanism critical for diversifying the human proteome. To systematically investigate its role in cell fate determination, we develop scCHyMErA-Seq, a scalable CRISPR-based exon deletion screening platform integrated with 10x Genomics single-cell transcriptomic readouts. This tool enables efficient exon deletion while simultaneously capturing Cas9/Cas12a guides and polyadenylated transcripts at single-cell resolution. Applying scCHyMErA-Seq to high-throughput profiling of alternative cassette exons, we identify numerous exons with pronounced regulatory effects on gene expression and cell cycle progression. Analysis of the alternative NRF1 exon-7 demonstrates that its inclusion modulates NRF1's regulatory function by influencing its recruitment to the promoters of target genes. Importantly, gene expression profiles generated using scCHyMErA-Seq accurately recapitulate findings from traditional, labor-intensive orthogonal methods, while offering enhanced scalability and efficiency. Overall, scCHyMErA-Seq represents a versatile platform for systematically unraveling the functional impact of alternative splicing by directly linking specific splicing variants to transcriptional phenotypes.

Alternative splicing is a fundamental regulatory mechanism that impacts nearly all protein-coding genes, modulating gene expression levels and generating multiple protein isoforms from a single gene[1–3]. This process expands proteomic diversity and enables precise regulation of gene function in response to environmental cues, and cell- or tissue-specific requirements[4–7]. Dysregulation of alternative splicing is associated with a wide range of diseases and disorders, including cancer and autism, underscoring its importance in health and disease[8–11]. Despite its prevalence, the functional relevance of most splice variants remains poorly understood,

[1]RNA Biology Laboratory, Center for Cancer Research (CCR), National Cancer Institute (NCI), National Institutes of Health (NIH), Frederick, MD, USA. [2]Advanced Biomedical Computational Science, Frederick National Laboratory for Cancer Research, Frederick, MD, USA. [3]Molecular Targets Program, Center for Cancer Research (CCR), National Cancer Institute (NCI), National Institutes of Health (NIH), Frederick, MD, USA. [4]Office of Science & Technology Resources, Center for Cancer Research (CCR), National Cancer Institute (NCI), National Institutes of Health (NIH), Bethesda, MD, USA. [5]Sequencing Facility Bioinformatics Group, Advanced Biomedical and Computational Science, Frederick National Laboratory for Cancer Research, Frederick, MD, USA. [6]These authors contributed equally: Bandana Kumari, Arun Prasath Damodaran, Wilfried M. Guiblet. ✉e-mail: michael.aregger@nih.gov; thomas.gonatopoulos@nih.gov

representing a major challenge in RNA biology and biomedical research[12].

To enable exon-resolution functional genomics, we previously developed CHyMErA (CRISPR Hybrid for Multiplexed Editing and Screening Applications), a combinatorial CRISPR tool that co-expresses Cas9 and Cas12a nucleases along with a hybrid guide RNA (hgRNA)[13,14]. This hgRNA, transcribed by RNA polymerase III (Pol III) from a single U6 promoter, fuses Cas9 and Cas12a gRNA sequences into a single transcript (Fig. 1a, b). Cas12a's intrinsic RNase activity[15,16] processes the hgRNA into distinct gRNAs, enabling combinatorial genetic perturbations, such as precise gene segment deletions[13,14]. In standard CRISPR gene knockout approaches, single-guide targeting of exonic sequences typically generates indels that produce either in-frame hypomorphic alleles or frameshifts that disrupt downstream coding sequence, complicating the interpretation of exon-specific functions. In contrast, CHyMErA directs Cas9 and Cas12a to flanking intronic regions, enabling precise exon excision while preserving surrounding coding sequences and, when applicable, the reading frame[13,17] (Fig. 1a). This strategy allows direct assessment of individual exon contributions to protein function.

We recently optimized CHyMErA by incorporating a modified *Acidaminococcus sp.* (*As*)Cas12a variant, substantially improving editing efficiency for large-scale exon deletion screens[17]. Using this approach, we identified hundreds of exons that affect cell fitness in human cells[17]. However, while fitness phenotypes are quantitative, they offer limited mechanistic insight into exon function. Moreover, fitness-altering exons are enriched in genes involved in gene expression regulation[17], highlighting the need for approaches that systematically link exon perturbations to complex molecular phenotypes, such as transcriptional programs.

Recent advances in genome editing and single-cell technologies have enabled pooled genetic perturbations with single-cell RNA sequencing (scRNA-Seq)[18–22]. These methods capture polyadenylated transcripts while identifying Cas9 gRNAs either through barcode-based strategies[18–21] or by directly sequencing gRNAs embedded within polyadenylated transcripts expressed from CROP-Seq vectors[22]. Despite their utility, these approaches face limitations, including

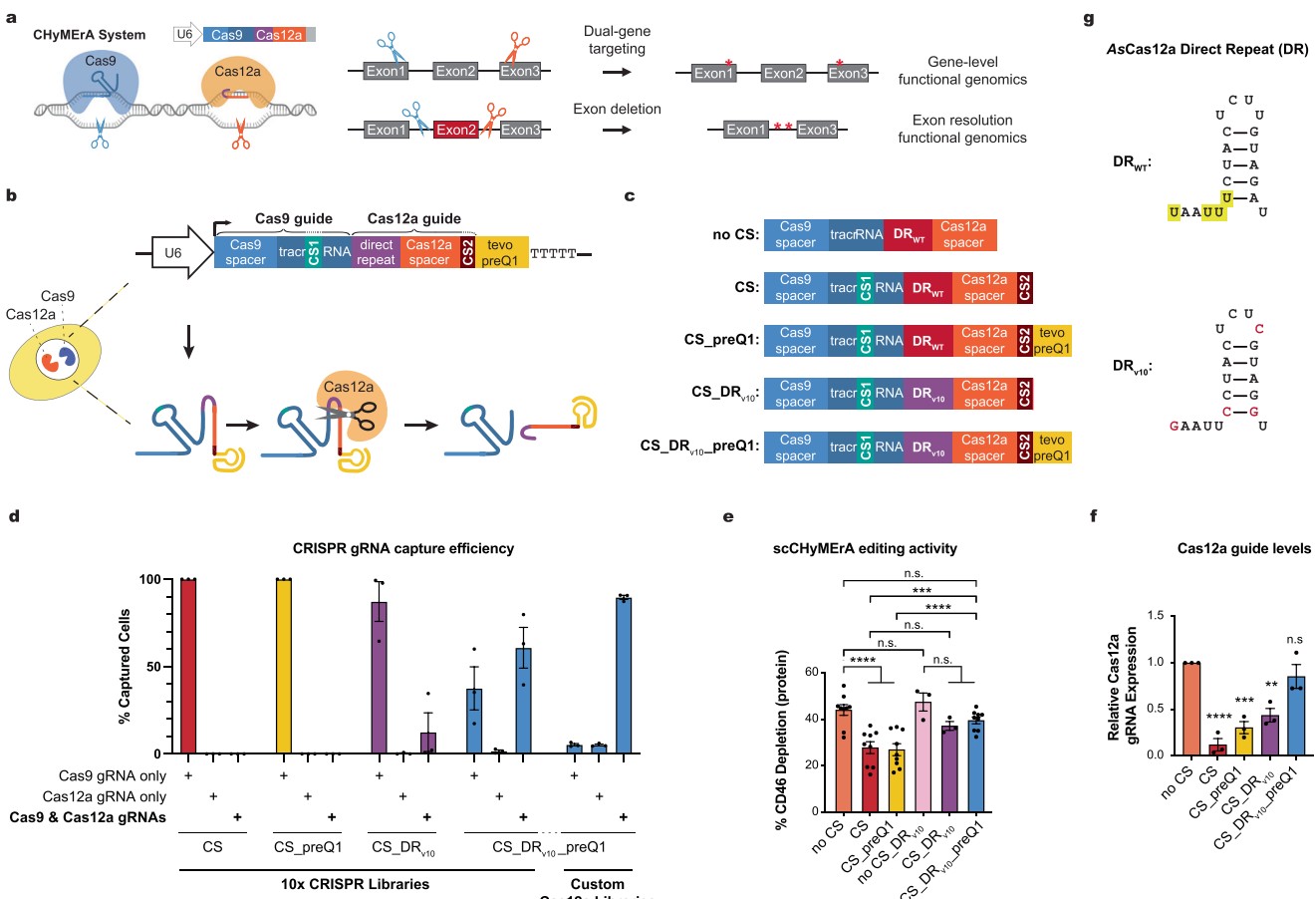

**Fig. 1 | Development of scCHyMErA-Seq, a combinatorial CRISPR platform integrated with single-cell transcriptomics. a** Schematic overview of the CHyMErA system illustrating dual-gene targeting and precise exon deletion. Graphical elements were created in BioRender. **b, c** Overview of the CHyMErA platform and the development of scCHyMErA-Seq. **b** Diagram of the CHyMErA system enabling multiplexed combinatorial CRISPR screening using Cas9 and Cas12a, including elements added for single-cell profiling. **c** Modifications introduced to enable efficient, simultaneous detection of Cas9 and Cas12a guide (g)RNAs in single cells, including capture sequences (CS), the direct repeat variant 10 (DR$_{v10}$), and the incorporation of the tevopreQ1 riboswitch. **d** Bar plots showing the proportions of cells with detectable Cas9 and/or Cas12a gRNAs using 10x Genomics droplet-based single-cell profiling. Different hgRNA designs (Fig. 1c) were tested using a standard 10x CRISPR library or a custom Cas12a library preparation workflow. Data are normalized to the total number of cells with detectable gRNAs (see Supplementary Data 1). Data represent mean ± SEM from three biological replicates. **e** *CD46* exon-3 deletion measured by flow cytometry in HAP1 cells transduced with three independent spacer sequence pairs, using the indicated hgRNA designs. Data represent mean ± SEM from biological replicates ($n = 3$–$9$). ****$p < 0.0001$, ***$p < 0.001$, n.s. not significant; two-way ANOVA. **f** Bar plots quantifying the relative expression levels of unprocessed hgRNAs and processed Cas12a gRNAs across different hgRNA designs assessed by northern blotting (see Supplementary Fig. 2a). Data represent mean ± SEM from three biological replicates. ****$p < 0.0001$, ***$p < 0.001$, **$p < 0.01$, n.s. not significant; one-way ANOVA. **g** Comparison of *As*Cas12a DR sequences. The wild-type (WT) DR is shown with the T-rich region highlighted in yellow; DR variant 10 (v10) is shown below, with nucleotide substitutions indicated in red.

barcode–guide uncoupling and inefficient gRNA detection in some CROP-Seq implementations[23–26], although targeted amplification of sgRNAs can partially mitigate the latter[24]. More recently, direct-capture strategies incorporating capture sequences into the tracrRNA have enabled robust detection of Cas9 gRNAs alongside polyadenylated transcripts[27–29]. However, equivalent direct-capture solutions for Cas12a gRNAs are lacking, and no analogous framework exists for exon-centric perturbation screens.

Here, we present single-cell CHyMErA sequencing (scCHyMErA-Seq), a platform that integrates high-throughput exon deletions with single-cell transcriptomics for high-resolution phenotypic profiling. By modifying the Cas12a direct repeat (DR) sequence and implementing a customized cDNA amplification protocol, we enable robust capture of Cas12a gRNAs within standard 10x Genomics workflows while improving editing efficiency. Using scCHyMErA-Seq, we interrogated 224 alternative exons and identified numerous exons that regulate transcriptional programs or whose perturbation alters cell-cycle dynamics. Among these, *NRF1* exon-7 was validated to influence transcription and promoter recruitment using targeted RNA-Seq and ChIP-Seq. Overall, scCHyMErA-Seq provides a framework for systematically mapping transcriptional consequences of exon-level perturbations at single-cell resolution.

## Results

### Scalable exon deletion tool for single-cell transcriptomics

To enable exon perturbation screens with single-cell resolution, we combined CHyMErA with 10x Genomics technology, which incorporates a capture sequence (CS1) into the Cas9 gRNA scaffold to facilitate guide RNA detection[27,28]. In combinatorial genetic screens such as CHyMErA, accurate detection of all expressed gRNAs within a perturbation, not only the Cas9 gRNA, is essential. Template switching during library cloning and lentiviral packaging can lead to gRNA decoupling[23–26], and individual gRNAs may pair with multiple partners, necessitating detection of both guides to unambiguously define each perturbation. Despite these needs, Cas12a gRNA capture in 10x Genomics single-cell RNA-Seq workflows has not been demonstrated.

We hypothesized that appending a distinct capture sequence (CS2) to the 3′ end of the Cas12a gRNA would enable its detection (Fig. 1b, c). However, when tested in HAP1 cells expressing *Sp*Cas9 and opCas12a[17,30], this design (CS) results in efficient Cas9 gRNAs capture but fails to detect Cas12a gRNAs, yielding 0% capture efficiency across thousands of cells (red bar, Fig. 1d). Furthermore, CS incorporation downstream of the Cas12a gRNA markedly reduces both CHyMErA's editing efficiency (red bar, Fig. 1e and Supplementary Fig. 1a, b), and Cas12a gRNA abundance (red bar, Fig. 1f and Supplementary Fig. 1c, d), regardless of the capture sequence used (Supplementary Fig. 1a–d). These observations underscore the need for alternative strategies to enable Cas12a gRNA capture while preserving CHyMErA activity.

We first considered whether CS2 is susceptible to exonucleolytic degradation. To stabilize the hgRNA, we incorporated the tevopreQ1 riboswitch[31] at its 3′ end, generating CS_preQ1 (Fig. 1c). While tevopreQ1 increases Cas12a gRNA abundance (yellow bars, Fig. 1f and Supplementary Fig. 2a, b), it fails to restore editing efficiency or Cas12a gRNA capture (yellow bars, Fig. 1d, e).

Next, we hypothesized that extending the hgRNA transcript with CS compromises Pol III processivity, leading to premature transcription termination and reduced Cas12a gRNA expression. Pol III premature transcription termination would eliminate the 3′ terminal $UUU_{OH}$ motif required for La binding, which stabilizes Pol III transcripts and promotes ribonucleoprotein assembly[32,33]. Consistent with this model, CS incorporation reduces Cas12a gRNA expression (red bar, Fig. 1f and Supplementary Figs. 1c, d and 2a). Notably, the Cas12a DR contains a U-rich stretch (Fig. 1g) that may act as a non-canonical Pol III termination sequence[34]. We therefore replaced the wild-type DR ($DR_{WT}$) with a previously validated variant containing fewer U residues

($DR_{v10}$)[35], generating CS_$DR_{v10}$ (Fig. 1c). This modification substantially increases Cas12a gRNA abundance relative to CS_$DR_{WT}$ (purple bars, Fig. 1f and Supplementary Fig. 2a, c).

Strikingly, $DR_{v10}$ also rescues CHyMErA exon deletion efficiency compromised by CS incorporation (purple bar, Fig. 1e) and enables detectable Cas12a gRNA capture, with ~12% of cells harboring both Cas9 and Cas12a gRNAs (purple bar, Fig. 1d). Incorporation of tevo-preQ1 into $DR_{v10}$ constructs (CS_$DR_{v10}$_preQ1; Fig. 1c) further increases dual Cas9 and Cas12a guide detection to 61% of cells (blue bars, Fig. 1d), albeit without additional gains in editing efficiency (blue bar, Fig. 1e).

Mechanistically, $DR_{v10}$ increases unprocessed hgRNA levels while reducing processed Cas9 gRNA abundance in the absence of Cas12a nuclease (Supplementary Figs. 1c and 2a), consistent with reduced premature Pol III termination and enhanced full-length hgRNA production. Importantly, these effects are specific to hgRNA transcribed by RNA Pol III: in hgRNAs expressed from an inducible RNA Pol II promoter (Supplementary Fig. 3a), the $DR_{v10}$ improves neither exon deletion efficiency nor Cas12a gRNA capture (Supplementary Fig. 3a–d). Together, these results indicate that Pol III-specific constraints on Cas12a gRNA expression can be alleviated through DR sequence optimization, and that combining $DR_{v10}$ with tevopreQ1 provides a robust strategy to enhance Cas12a guide expression and capture.

Despite these improvements, Cas12a gRNA detection remained suboptimal (i.e., ~61%). To further enhance capture, we introduced a targeted amplification step during 10x Genomics library preparation using a custom oligo annealing to $DR_{v10}$, enabling selective amplification of Cas12a-derived cDNAs (Supplementary Fig. 4a, b; Methods). This modification increases Cas12a gRNA capture to ~90% ("Custom Cas12a Libraries", Fig. 1c and Supplementary Fig. 4c). Across all profiled cells, including those lacking any detectable gRNAs, this approach improves dual-guide detection from 47 to 72% (Supplementary Fig. 4d and Supplementary Data 1). Collectively, optimization of the hgRNA scaffold and library preparation workflow substantially enhances CHyMErA performance in single-cell exon deletion assays.

This optimized platform enables efficient combinatorial genetic perturbations and precise gene segment deletions while allowing simultaneous capture of Cas9 and Cas12a gRNAs together with polyadenylated transcripts at single-cell resolution. We term this approach single-cell CHyMErA sequencing (scCHyMErA-Seq).

### Profiling alternative cassette exons using scCHyMErA-Seq

To validate scCHyMErA-Seq, we conducted a screen targeting the deletion of 224 alternative cassette exons from 161 genes. These exons were previously identified as influencing cell fitness[17]. We focused on frame-preserving exons predicted to generate alternative protein variants[17], and prioritized those within genes implicated in transcriptional phenotypes[28]. We constructed a lentiviral library comprising 1066 hgRNAs, with each exon targeted by three independent guide pairs directing Cas9 and Cas12a to flanking intronic regions (Fig. 2a). Guides were selected from our prior large-scale exon deletion screen[17], based on high on-target activity, minimal predicted off-target effects, and deletion sizes below 2 kb. In addition to exon deletions, the library includes two independent hgRNAs for gene knockouts, with both Cas9 and Cas12a nucleases inducing mutations in the targeted genes (Fig. 2a). This dual-nuclease strategy enhances gene inactivation efficiency[13,17]. Knockout guides were randomly selected from our genome-wide CHyMErA gene knockout library[36], which is filtered for high on-target activity, minimal off-target potential, and coverage across all transcript isoforms, with Cas9 and Cas12a guides targeting distinct regions of each gene. The scCHyMErA-Seq library also contains 40 intergenic and 40 non-targeting hgRNAs as negative controls (Supplementary Data 2).

HAP1 cells expressing Cas9 and Cas12a were transduced at a low multiplicity of infection (<0.1), and 4 days later, subjected to droplet-based 3′ scRNA-Seq using the 10x Genomics platform (Fig. 2b). Illumina

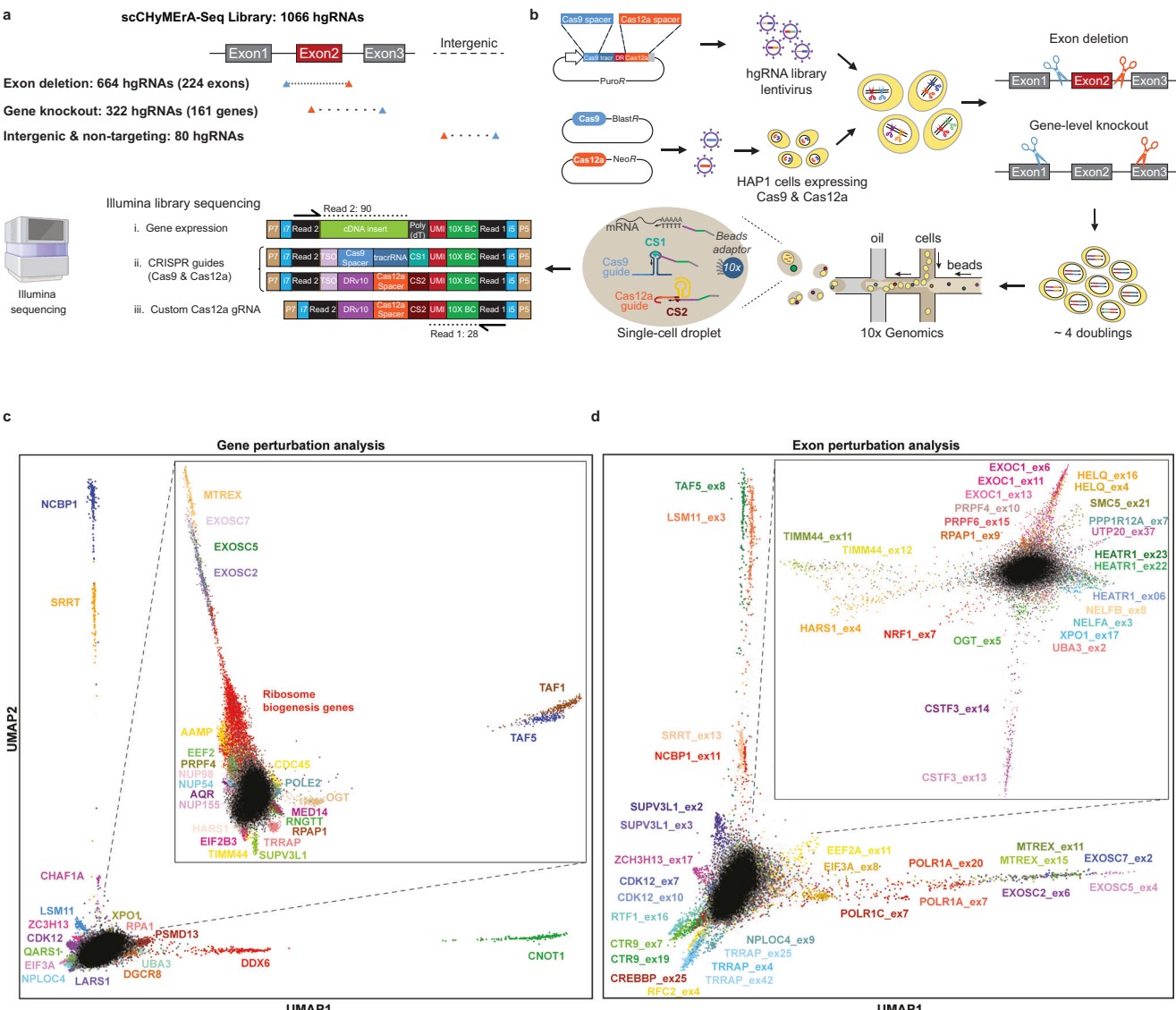

**Fig. 2 | Application of scCHyMErA-Seq for the targeted deletion of 224 alternative cassette exons. a** Schematic representation of the lentiviral scCHyMErA-Seq library targeting 224 alternative cassette exons for deletion and 161 genes for knockout. Blue and orange triangles indicate Cas9 and Cas12a target sites, respectively, for each hgRNA category. **b** Overview of the scCHyMErA-Seq experimental pipeline. Graphical elements were created in BioRender. **c**, **d** UMAP visualization of HAP1 cells expressing hgRNAs inducing gene-level knockouts (**c**) or exon deletions (**d**) following linear discriminant analysis (LDA). Insets show UMAPs recalculated after removal of cells associated with the highlighted perturbations in the original plots. Distinct clusters are labeled by the targeted gene or exon. Black dots denote cells carrying hgRNAs targeting other genes, exons, or intergenic controls.

sequencing enabled the linkage of gene expression profiles to guide identities, yielding on average of >47,000 polyadenylated transcript reads and ~9000 gRNA reads per cell.

We profiled 412,865 cells, of which 92% contain detectable Cas9 and Cas12a gRNAs, while fewer than 3% lack any gRNA detection (Supplementary Fig. 5a). The integration of CRISPR screens with scRNA-Seq provides the additional advantage of using gRNA sequences as unique barcodes, facilitating the accurate identification and removal of cell doublets. After filtering out low-quality cells and doublets—an artifact commonly associated with the high-throughput 10x Genomics platform—we retained 212,217 high-quality and ~confidence single cells (51.4%) for downstream analysis (Supplementary Fig. 5b–i). Cells carrying gene knockouts, exons deletions, or control hgRNAs exhibit comparable UMI counts, number of detected genes, and mitochondrial transcript fractions, indicating minimal perturbation-induced bias in overall cell state or data quality (Supplementary Fig. 5j–l). On average, we obtained ~200 cells per hgRNA,

~600 cells per targeted exon, ~400 cells per gene-level knockout, and ~16,000 cells corresponding to control guides.

To visualize the data and reduce their complexity, we applied dimensionality reduction techniques, specifically the uniform manifold approximation and projection (UMAP). Furthermore, to enhance signal-to-noise, we applied Mixscape to exclude unperturbed cells[37,38], and performed Leiden clustering[39] on gene- and exon-level perturbations separately. Perturbations affecting genes within the same complex or pathway are expected to induce correlated transcriptional responses and group closely together. Consistently, cells with knockouts of RNA exosome components (*EXOSC2*, *EXOSC7*, and *MTREX*) form a distinct cluster, as do cells with *TAF1* and *TAF5* knockouts, both members of the TFIID complex (Supplementary Fig. 6a, b). At the exon level, deletions in *RTF1* and *CTR9*—components of the PAF1 complex, a regulator of RNA polymerase II elongation[40]—are enriched in specific clusters (Supplementary Fig. 6c, d). Despite the intrinsic noise of single-cell data in cell lines, these results demonstrate reproducible

transcriptional signatures driven by both gene- and exon-level perturbations.

To further maximize separation of perturbations, we applied linear discriminant analysis (LDA) followed by UMAP visualization (Fig. 2c, d). This approach improved resolution of both gene- and exon-specific transcriptional phenotypes. For example, knockout cells targeting nuclear cap-binding complex-related genes *NCBP1* and *SRRT*[41] cluster in close proximity (Fig. 2c). Similarly, cells lacking genes associated with the SSU processome, a key regulator of rRNA transcription, processing, assembly, and ribosomal subunit maturation[42], form a coherent cluster. This includes members of the UTP-A (*WDR43* and *HEATR1*), UTP-B (*DDX21*), UTP-C (*NOL6*), and ANN (*NGDN*, *NOL10*) subcomplexes, as well as additional ribosome biogenesis factors (*NOP14*, *UTP20*, *DIMT1*, *POP1*, *DHX33*, and *RRP12*) (Fig. 2c and Supplementary Fig. 6e).

At the exon level, we observed distinct clusters driven by deletions of individual alternative cassette exons, indicating that exon-specific perturbations can elicit unique transcriptional responses. Deletion of *TAF5* alternative exon-8, which regulates TFIID assembly, produces a discrete cluster (Fig. 2d), consistent with its previously described role in gene regulation[17]. Similar clustering patterns are observed for alternative exons in *NCBP1* and *SRRT*[41], as well as in *CTR9* and *RTF1* (Fig. 2d). Notably, deletion of individual *CTR9* exons generates distinct but closely positioned clusters, highlighting the ability of scCHyMErA-Seq to resolve transcriptional phenotypes arising from the loss of specific alternative exons within the same gene.

## Alternative exons frequently regulate transcriptional phenotypes

We next leveraged our dataset to identify genes and exons with significant transcriptomic effects by aggregating cells harboring distinct hgRNA-defined perturbations. Comparing cells expressing intergenic versus non-targeting hgRNAs revealed highly similar gene expression profiles, indicating that intergenic guides provide an appropriate baseline for downstream analyses (Supplementary Fig. 7a). Differential gene expression analyses of cells expressing individual hgRNAs versus intergenic controls demonstrate consistent transcriptional responses across independent hgRNAs targeting the same exon (or gene), highlighting strong reproducibility (Supplementary Fig. 7b, c).

Cells expressing the same hgRNA were grouped, treating distinct hgRNAs targeting the same gene or exon as biological replicates (two for gene knockouts and three for exon deletions). Comparisons to 40 intergenic control groups using DESeq2 revealed that 103 of the 161 genes (64%) and 101 of the 224 exons (45%) significantly alter the expression of at least 200 genes (Supplementary Figs. 8 and 9 and Supplementary Data 3 and 4). Genes with the most pronounced effects include *CNOT1*, *NCBP1*, and *TRRAP*, aligning with their roles in gene expression regulation and RNA stability (Supplementary Fig. 8 and Supplementary Data 3). Exons driving widespread gene expression changes include *TAF5* exon-8 (1908 differentially expressed genes), *CREBBP* exon-25 (2184 genes), *EXOSC5* exon-4 (844 genes), and *NRF1* exon-7 (386 genes) (Supplementary Fig. 9 and Supplementary Data 4). Perturbations enriched distinct biological processes, including metabolic pathways, stress responses, and signaling cascades (Supplementary Fig. 10), demonstrating scCHyMErA-Seq's capacity to assess the transcriptomic impact of hundreds of individual exons in a single experiment.

Next, we conducted unbiased clustering of exon- or gene-level perturbations, correlating them based on the expression profiles of highly variable genes. This analysis revealed distinct clusters of exons and associated genes enriched for functional categories such as ribosome biogenesis, protein synthesis, and RNA decay (Fig. 3a and Supplementary Fig. 11a, b). For example, *TAF5* exon-8 and *LSM11* exon-3 perturbations produce positively correlated changes in gene

expression (Fig. 3a), consistent with their close clustering in UMAP-LDA analyses (Fig. 2d).

*LSM11* encodes an Sm-like protein within the U7 small nuclear ribonucleoprotein (snRNP), a key regulator of replication-dependent histone pre-mRNA processing. Exon-3 of *LSM11* overlaps the Sm domain, likely affecting its ability to bind U7 snRNA and impairing U7 snRNP function. Proper histone RNA processing by U7 snRNP prevents polyadenylation, making transcripts undetectable with oligo(dT) primed cDNA. Deletion of *LSM11* exon-3 leads to increased detection of replication-dependent histone transcripts, consistent with impaired U7 snRNP activity (Fig. 3b). Similarly, *TAF5* exon-8 deletion results in increased histone transcript expression, suggesting a shared role in histone 3′ end processing (Fig. 3b). Targeted deletion of these exons with the same three hgRNAs confirmed efficient exon skipping (Fig. 3c and Supplementary Fig. 12a) and recapitulated the observed increase in polyadenylated histone transcripts (Fig. 3d). These results were further validated in HEK293 Flp-In cells engineered to express either full-length TAF5 or the TAF5-Δex8 isoform following siRNA-mediated depletion of endogenous TAF5 (Supplementary Fig. 12b, c).

Similar to the well-established function of LSM11 in histone mRNA processing, our data suggest that TAF5 also plays a critical role in histone regulation. TAF5 and other TAF proteins have been implicated in promoting expression of RNA Pol II-transcribed U snRNAs[43], such as U7, providing a potential mechanistic link between *TAF5* exon-8 and histone pre-mRNA processing. In summary, scCHyMErA-Seq uncovered a previously unrecognized role for *TAF5* exon-8 in histone mRNA regulation and demonstrated that deletion of exons within the same complex or pathway often produces clustered transcriptional profiles, offering insight into exon-specific functions.

## NRF1 exon-7 is critical for its transcriptional output

One cassette exon identified as influencing transcriptional programs is alternative exon-7 of *NRF1*, an essential transcription factor that regulates nuclear genes involved in cellular energetics by binding GC-rich promoter elements[44]. Exon-7 encodes a 66–amino acid segment that partially overlaps the NRF1 DNA-binding domain (Fig. 4a). scCHyMErA-Seq identified 373 differentially expressed genes following exon-7 deletion, with strong enrichment for NRF1 consensus motifs in their promoters (Fig. 4b, Supplementary Fig. 13a, and Supplementary Data 4).

To validate these observations, we cloned the three independent library hgRNAs targeting *NRF1* exon-7 and confirmed efficient exon skipping at both RNA and protein levels (Fig. 4c and Supplementary Fig. 13b). We then examined expression of three scCHyMErA-Seq–identified NRF1 target genes—*PUDP*, *ANKRD26*, and *YEATS4*. Deletion of exon-7 consistently reduces *PUDP* and *YEATS4* expression across all hgRNAs (Fig. 4d). For *ANKRD26*, only one hgRNA reached statistical significance, although all three showed concordant directional effects (Fig. 4d). Notably, the hgRNA with the strongest transcriptional impact also exhibits the highest exon-skipping efficiency and greatest expression of the Δexon-7 NRF1 isoform (Fig. 4c).

To further assess the robustness and sensitivity of scCHyMErA-Seq, we employed two complementary exon perturbation strategies coupled with bulk RNA-Seq. First, we generated stable HEK293 Flp-In cell lines expressing either full-length (FL) or exon-7-deleted (ΔE7) NRF1 isoforms in a doxycycline-inducible manner, following siRNA-mediated depletion of endogenous NRF1 (Fig. 4e). Bulk RNA-Seq analysis identified 800 genes differentially expressed between the FL- and ΔE7-NRF1 isoforms, confirming the functional importance of exon-7 (Fig. 4f and Supplementary Data 5). Second, we induced exon-7 skipping using adenine or cytosine base editors targeting the exon's splice donor or acceptor sites. Reverse transcriptase (RT)-PCR and western blot analyses confirmed elevated expression of the ΔE7-NRF1 isoform in cells transfected with these constructs (Supplementary Fig.

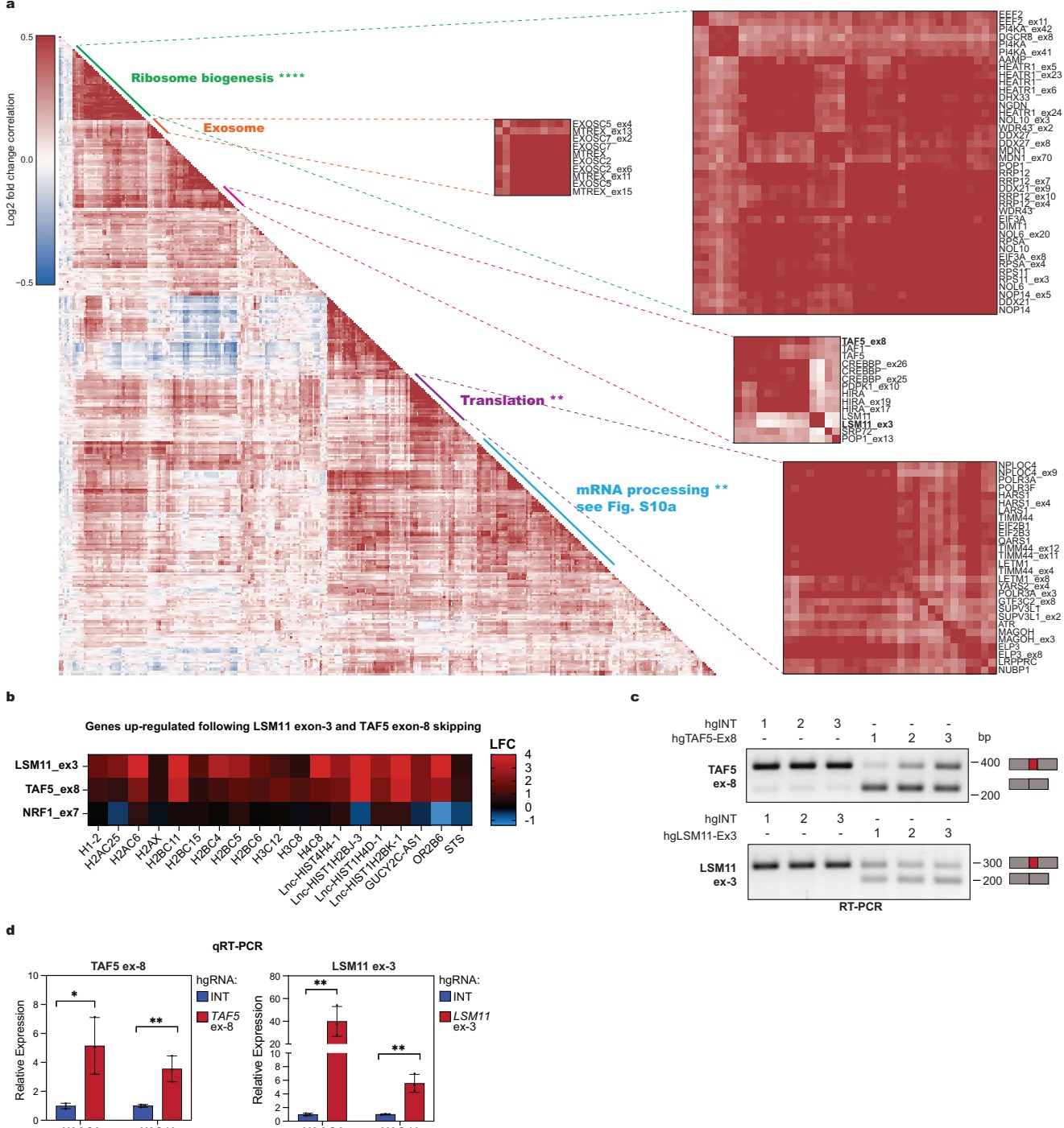

**Fig. 3 | scCHyMErA-Seq reveals correlated gene expression phenotypes linked to distinct molecular pathways from exon- and gene-level perturbations.**
**a** Heatmap of pairwise correlations between log2 fold-change values of differentially expressed genes across gene knockout and exon deletion (_ex) perturbations identified by scCHyMErA-Seq. Distinct sub-clusters are annotated with Molecular Signatures Database (MSigDB) pathway terms. Adjusted $p$ values for pathway enrichment are indicated; ****$p < 0.0001$, **$p < 0.01$; one-sided Fisher's exact test corrected for multiple hypothesis (Benjamini–Hochberg adjusted). Representative genes from each subcluster are listed on the right. **b** Heatmap depicting gene expression changes in replication-dependent histones across the indicated perturbed exons. All the genes that are commonly upregulated

following *LSM11* exon-3 and *TAF5* exon-8 deletions are shown. *NRF1* exon-7 deletion serves as a control. **c** RT-PCR analysis of TAF5 exon-8 (top) and LSM11 exon-3 (bottom) inclusion in RNA from HAP1 cells transduced with three independent intergenic hgRNA control sequences or three independent hgRNA sequences targeting *TAF5* exon-8 or *LSM11* exon-3 for deletion. Constructs compatible with the scCHyMErA-Seq system were used for exon deletion. **d** qRT-PCR analysis of polyadenylated histone gene transcripts following deletion of *TAF5* exon-8 or *LSM11* exon-3 using the three independent intergenic hgRNA controls or exon-targeting hgRNAs shown in Fig. 3c. Data represent mean ± SD from three biological replicates; **$p < 0.01$, *$p < 0.05$, two-tailed unpaired $t$-test.

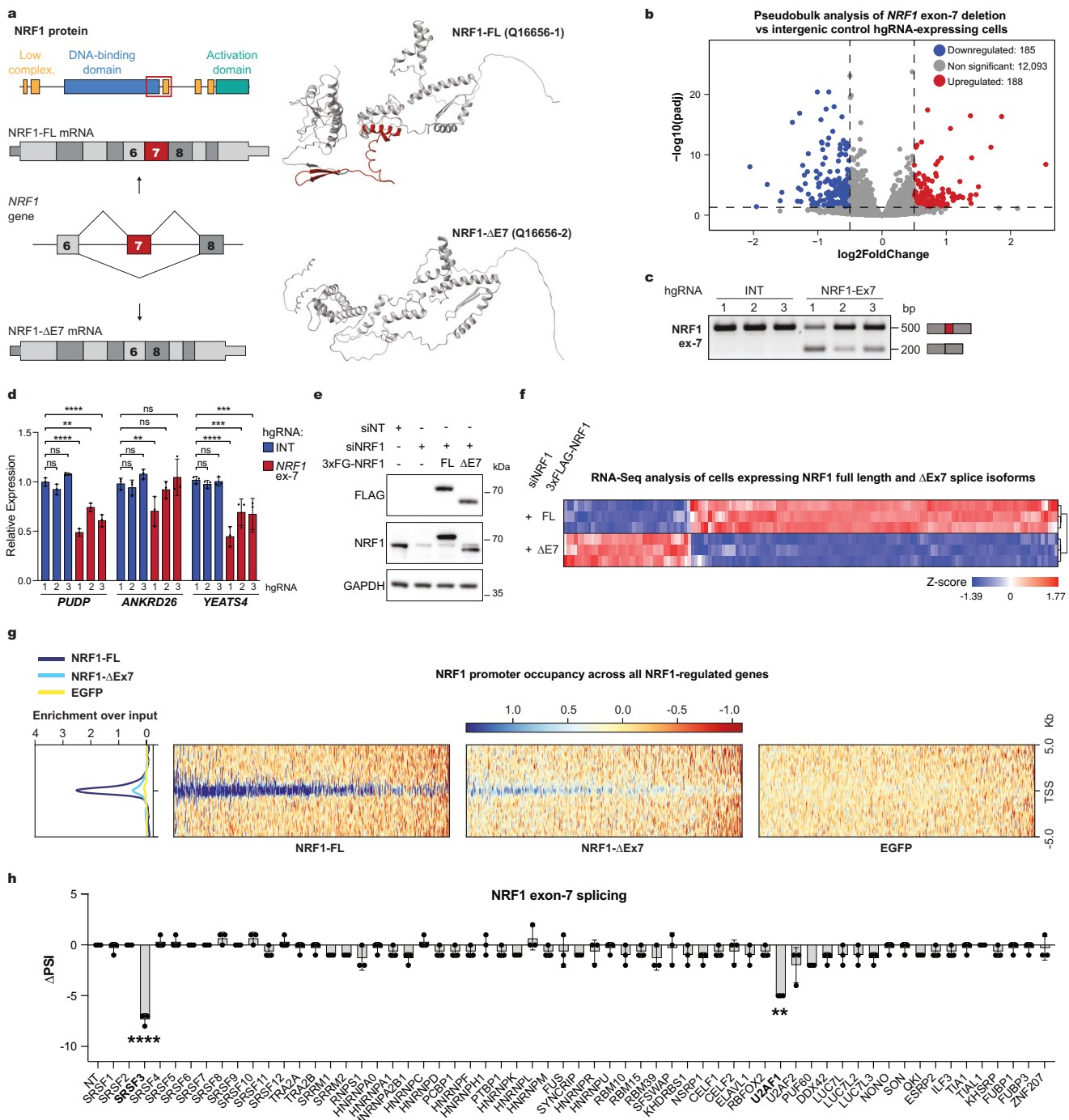

**Fig. 4 | scCHyMErA-seq reveals exon-7 as essential for NRF1 transcriptional function. a** Schematic of NRF1 protein domains and exon structure. The protein diagram (top) highlights low-complexity regions (orange), the DNA-binding domain (blue), and the transcriptional activation domain (turquoise). The splicing schematic (middle) depicts the full-length (FL, top) and exon-7–deleted (ΔE7, bottom) mRNA isoforms. AlphaFold structural predictions[86,87] for the NRF1 full-length (top right) and Δexon-7 (bottom right) isoforms are shown, with the exon-7-encoded region highlighted in red. **b** Volcano plot of DESeq2 pseudobulk analysis comparing cells expressing *NRF1* exon-7–targeting hgRNAs with intergenic controls. Significantly upregulated (red) and downregulated (blue) genes are indicated (adjusted $p < 0.05$ and |log2 fold-change|>0.5). $p$ values were computed using DESeq2's Wald test with Benjamini–Hochberg correction. **c** RT-PCR analysis of NRF1 exon-7 inclusion in RNA from HAP1 cells transduced with three independent intergenic control hgRNAs or three independent hgRNAs targeting *NRF1* exon-7 for deletion using scCHyMErA-Seq-compatible constructs. Representative images from three independent experiments. **d** qRT-PCR validation of *NRF1* exon-7-

dependent gene expression changes using the hgRNAs shown in Fig. 4c. Data represent mean ± SD from three biological replicates; ****$p < 0.0001$, ***$p < 0.001$, **$p < 0.01$; two-way ANOVA with Dunnett's multiple comparisons test. **e** Western blot analysis of NRF1 isoforms in HEK293 Flp-In cells stably expressing doxycycline-inducible 3×FLAG-tagged NRF1 with exon-7 included (FL) or excluded (ΔE7). The ectopic isoforms carry silent mutations at the siRNA target site, conferring resistance to siNRF1. Cells were transfected with control (siNT) or NRF1-targeting (siNRF1) siRNAs. Blots were probed for NRF1, FLAG, and GAPDH (loading control). Representative images from three independent experiments. **f** RNA-Seq analysis showing Z-score normalized expression changes of differentially expressed genes in siNRF1-treated cells rescued with NRF1-FL or -ΔE7 isoforms. **g** ChIP-Seq density profiles of NRF1-FL and -ΔE7 isoform enrichment over input at transcription start sites (TSS) ± 5 kb flanking regions of the genes regulated by NRF1 exon-7. **h** siRNA screen of 60 splicing regulators identifying factors controlling NRF1 exon-7 inclusion. Data represent the mean ± SD from three biological replicates; ****$p < 0.0001$, **$p < 0.01$; two-way ANOVA.

13c, d). Bulk RNA-Seq analysis of these cells further demonstrated widespread gene expression changes with 812 affected genes (Supplementary Fig. 13e and Supplementary Data 5). The two approaches show extensive overlap ($p = 5.9e-304$; Fisher's exact test) and strong correlation ($R = 0.74$) in gene expression changes (Supplementary Fig. 13f, g).

Deletion of exon-7 does not produce detectable changes in NRF1 subcellular localization or substantially affect its protein–protein interactions, as assessed by immunofluorescence and TurboID mass spectrometry, respectively (Supplementary Fig. 14a, b), suggesting that exon-7 does not regulate NRF1 spatial distribution or interactome.

Given the overlap of exon-7 with the NRF1 DNA-binding domain (Fig. 4a), we next examined its role in promoter recruitment. ChIP-Seq analysis comparing FL- and ΔE7-NRF1 revealed a pronounced reduction in NRF1 occupancy at gene promoters in the absence of exon-7 (Fig. 4g). This effect was particularly evident for NRF1-upregulated genes (Supplementary Fig. 15a), underscoring the specific importance of exon-7 in facilitating NRF1 binding to its target sites. Reduced binding at promoters of *PUDP*, *ANKRD26*, and *YEATS4* was further validated by ChIP-qPCR and qRT-PCR using independent samples (Supplementary Fig. 15b–d). These findings highlight the functional significance of exon-7 in NRF1's transcriptional function, suggesting that its deletion disrupts DNA binding and compromises NRF1's capacity to regulate gene expression programs effectively.

Finally, to identify regulators of exon-7 inclusion, we performed an siRNA screen targeting 60 splicing factors. Our results pinpointed SRSF3 as the key factor promoting endogenous exon-7 inclusion (Fig. 4h). The role of SRSF3 in regulating NRF1 exon-7 was validated through three independent siRNA treatments, as well as by partially rescuing NRF1 exon-7 splicing through the addition of an SRSF3 variant resistant to siRNA-mediated knockdown (Supplementary Fig. 16a–c). Together, these findings demonstrate that SRSF3-mediated inclusion of exon-7 is essential for NRF1's transcriptional function by controlling its recruitment to target gene promoters. Furthermore, our results underscore the utility of scCHyMErA-Seq as a powerful tool for systematically identifying exons that influence transcriptional phenotypes.

### scCHyMErA-Seq reveals exons influencing cell cycle distribution

Alternative splicing plays a central role in regulating cell fate decisions, including cell cycle progression and differentiation[45–48]. Therefore, we next leveraged the scCHyMErA-Seq dataset to systematically identify exons whose perturbation alters cell cycle phase distributions. Cell cycle phases were assigned to individual cells using Scanpy-based scoring of cell cycle marker genes[49], followed by aggregation of cells according to their genetic perturbation. As expected, cells transduced with intergenic or non-targeting hgRNAs exhibit comparable phase distributions, with approximately 17–18% of cells in G1 (Fig. 5a).

To identify perturbations affecting cell cycle progression, we grouped cells by hgRNA and quantified the fraction of cells in each phase. Independent hgRNAs targeting the same exon or gene were treated as biological replicates and compared against intergenic controls. This analysis identified 69 exons and 76 genes whose perturbation significantly alters cell cycle distributions (Fig. 5a, Supplementary Fig. 17a and Supplementary Data 6). Notably, several affected exons reside in genes previously implicated in cell cycle control, including *TRRAP*[60,51], *SKP2*[52], and *HEATR1*[53].

To validate these results, we cloned 33 hgRNAs targeting ten exons and their corresponding genes (two to three independent hgRNAs per exon and one per gene), along with two intergenic controls. Following transduction into HAP1 cells, exon deletion efficiency was assessed by RT-PCR, confirming efficient and specific exon skipping (Fig. 5b). Flow cytometry analysis after propidium iodide staining showed that perturbation of seven out of ten tested exons recapitulated the cell cycle phenotypes predicted by scCHyMErA-Seq (Fig. 5c). Notably, six of these seven exons reside in genes whose knockout

similarly affects cell cycle progression according to flow cytometry (Supplementary Fig. 17b). Together, these results demonstrate that scCHyMErA-Seq enables systematic identification of exons that influence cell cycle phase distributions, linking exon-level perturbations to specific cell states.

### Exon- and gene-level phenotypes can diverge

To assess exon-specific phenotypes that may deviate from corresponding gene-level effects, we examined the correlation of log2 fold-change values for differentially expressed genes across 224 exon deletions and their respective gene knockouts. As expected, exon deletions and gene knockouts show a strong overall concordance ($R = 0.65$), indicating broadly similar transcriptional effects (Fig. 5d).

We next performed differential expression analyses directly comparing each exon deletion with its corresponding gene knockout (Supplementary Fig. 18a and Supplementary Data 7). The largest discrepancies are observed for genes whose knockout elicits broad transcriptional changes, whereas deletion of individual exons generally results in more subtle effects, consistent with many exons being dispensable for gene function.

To systematically identify exons whose perturbation generates transcriptomic profiles distinct from gene knockout effects, we computed correlation coefficients between exon- and gene-level perturbations. Most exon deletions (124 of 224) are strongly correlated with their corresponding knockouts ($R > 0.4$), indicating that exon-level perturbations typically recapitulate gene-level phenotypes (Supplementary Data 7). However, a subset of exons displays weaker correlations despite affecting a large number of genes. A notable example is *SKP2* exon-6, whose deletion alters the regulation of a gene set distinct from that regulated by *SKP2* knockout (Fig. 5e, f). Genes uniquely affected by exon-6 deletion are enriched for cell cycle-related pathways (Fig. 5g).

*SKP2* encodes an E3 ubiquitin ligase that promotes degradation of cell cycle regulators such as p27 to facilitate G1–S transition[52]. Both *SKP2* knockout and exon-6 deletion increase the fraction of cells in G1-phase (Fig. 5c and Supplementary Fig. 17b), with a stronger effect observed following gene knockout, consistent with partial loss of SKP2 function upon exon deletion (Supplementary Fig. 18b). Exon-6 overlaps the leucine-rich repeat domain implicated in substrate recognition, suggesting that its removal impairs—but does not abolish—SKP2 activity (Supplementary Fig. 18c). Consistent with this interpretation, reanalysis of differential expression restricted to G1-phase cells reduces the differences between exon-6 deletion and gene knockout and largely eliminates enrichment for cell cycle-related pathways (Supplementary Fig. 18d, e). Together, these findings suggest that exon-level perturbations can fine-tune protein function, producing transcriptional outcomes that only partially overlap with gene knockouts and depend on cellular state.

Beyond *SKP2* exon-6, several additional exons exhibit stronger phenotypic effects upon deletion than the corresponding gene knockouts (Supplementary Fig. 18a), including *SMG5* exon-5, *NUBP1* exon-4, *TFAM* exon-5, and *ESPL1* exon-8, among others (Supplementary Fig. 19a–h). SMG5 is a core component of the nonsense-mediated decay (NMD) pathway[54,55] and has been linked to colorectal cancer[56]. Exon-5 overlaps a tetratricopeptide-like helical domain at the SMG5 N-terminus (Supplementary Fig. 19a, b), that mediates binding to SMG7[57], suggesting that exon-5 loss may impair SMG7 interaction and impact NMD efficiency. *NUBP1* encodes a cytosolic iron–sulfur cluster scaffold with ATPase activity essential for transfer of iron–sulfur clusters to target proteins[58]. Exon-4 overlaps the ATPase domain (Supplementary Fig. 19c, d), raising the possibility that it influences the timing or specificity of protein–protein interactions during iron–sulfur cluster delivery[59]. Finally, TFAM is a mitochondrial transcription factor and high-mobility group (HMG) protein that binds and unwinds mitochondrial DNA and recruits the mitochondrial RNA polymerase to promoters[60,61]. An exon-5-skipped TFAM isoform was identified

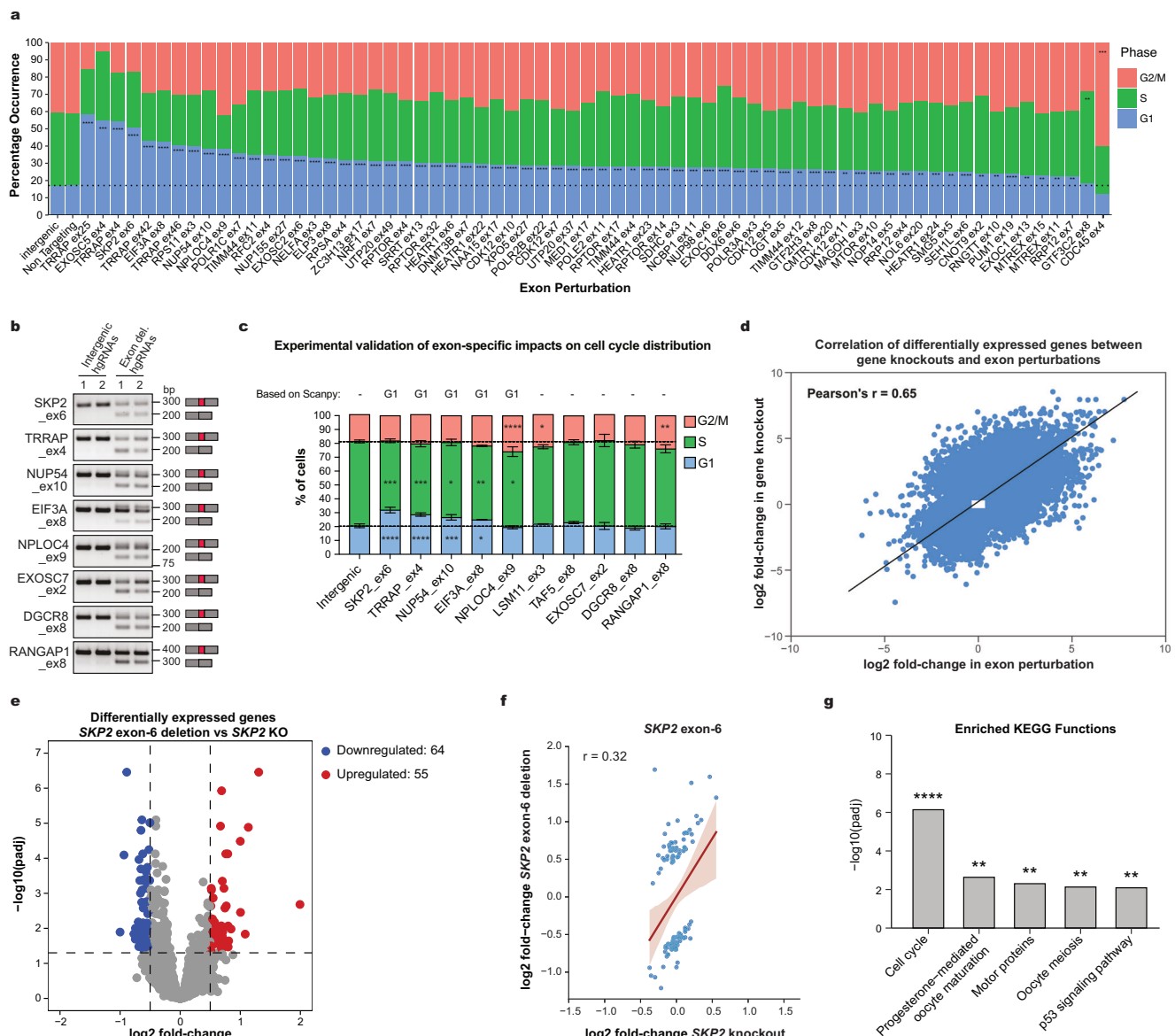

**Fig. 5 | scCHyMErA-Seq identifies cassette exons influencing cell cycle distribution. a** Stacked bar plot summarizing exons whose deletion affects cell cycle phase distribution, as identified by scCHyMErA-Seq. Phases with significantly increased cell fractions are indicated (****$p < 0.0001$, ***$p < 0.001$, **$p < 0.01$; One-sided Fisher's exact test with Benjamini–Hochberg correction). **b** RT-PCR analysis of exon inclusion for the indicated genes in RNA from HAP1 cells transduced with two independent intergenic control hgRNAs or two independent hgRNAs targeting the indicated exons using scCHyMErA-seq–compatible constructs. Single experiment performed. **c** Validation of scCHyMErA-Seq results by propidium iodide staining and flow cytometry. Stacked bar plots indicate the effect of exon deletions on cell cycle distribution; significantly increased phases are indicated. Data represent mean ± SD from three biological replicates, each calculated as the average of two independent hgRNA treatments; **$p < 0.01$, *$p < 0.05$; one-way ANOVA. **d** Scatter plot comparing log2 fold-change values of differentially expressed genes across all perturbations between gene knockouts and exon deletions. The two-sided Pearson's correlation coefficient is shown. **e** Volcano plot of DESeq2 pseudobulk analysis comparing *SKP2* exon-6 deletion with *SKP2* knockout. Significantly upregulated (red) and downregulated (blue) genes are indicated (adjusted $p < 0.05$, |log2 fold-change|> 0.5). $p$ values were computed using DESeq2's Wald test with Benjamini–Hochberg correction. **f** Scatter plot comparing log2 fold-change values of genes between *SKP2* knockout versus intergenic controls (x-axis) and *SKP2* exon-6 deletion versus intergenic controls (y-axis). Only genes differentially expressed between *SKP2* knockout and exon-6 deletion are shown. **g** KEGG pathway enrichment analysis of differentially expressed genes between *SKP2* knockout and *SKP2* exon-6 deletion. Pathways with $p$adj <0.01 (****$p < 0.0001$, ***$p < 0.001$; Fisher's exact one-tailed test with multiple-testing correction using g:Profiler's g:SCS method); all expressed genes were used as background.

previously[62], and this exon partially overlaps the second HMG domain, suggesting a role in modulating TFAM–DNA-binding affinity (Supplementary Fig. 19e, f). Indeed, genes affected by *TFAM* exon-5 deletion are strongly enriched for mitochondrial pathways related to oxidative phosphorylation. Notably, a previous study identified an ERK-dependent phosphorylation site within exon-5 that reduces TFAM-mediated mitochondrial DNA transcription with implications for Parkinson's disease[63]. Together, these vignettes illustrate how

scCHyMErA-Seq enables future hypothesis-driven prioritization of splice isoforms for mechanistic studies in health and disease.

**Comparing scCHyMErA-Seq with orthogonal methodologies**
To assess reproducibility and benchmark scCHyMErA-Seq performance, we initially conducted an independent single-cell exon deletion experiment using a focused library comprising nine intergenic hgRNAs and four hgRNAs targeting *NRF1* exon-7 for deletion. Data

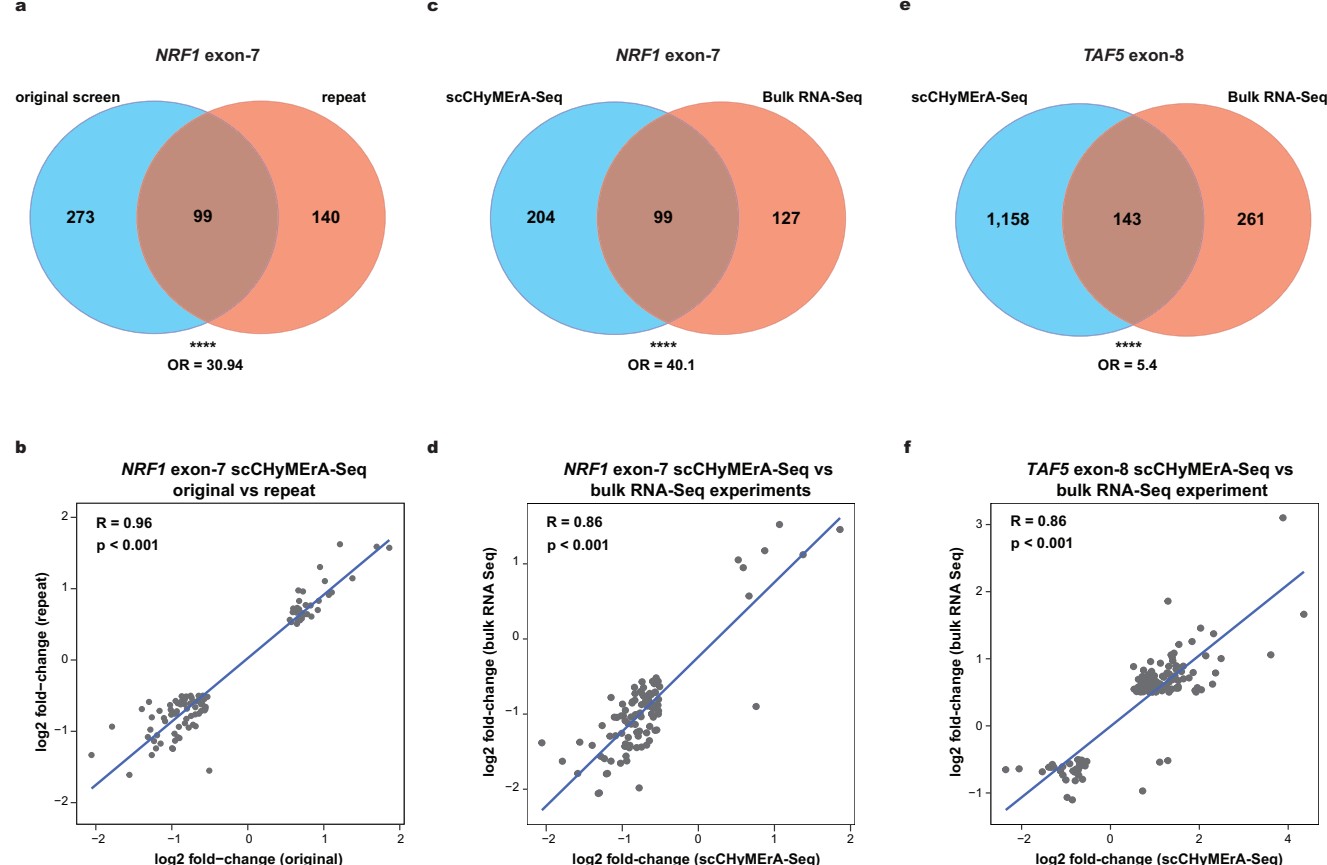

**Fig. 6 | scCHyMErA-Seq enables high-throughput and accurate identification of exons and genes that modulate transcriptional phenotypes. a**, **c**, **e** Venn diagrams showing the overlap of differentially expressed genes regulated by perturbation of *NRF1* exon-7 (**a**, **c**) or *TAF5* exon-8 (**e**). Overlaps are compared between our original scCHyMErA-Seq screen in HAP1 cells and either a small-scale replicate screen in HAP1 cells (**a**) or bulk RNA-Seq in HEK293 cells following isoform rescue experiments (**c**, **e**). Statistical significance was determined by two-sided Fisher's exact test (****$p < 0.0001$), and the corresponding odds ratios (OR) are indicated. **b**, **d**, **f** Scatterplots showing the Pearson correlation of log2 fold-changes for differentially expressed genes identified by scCHyMErA-Seq and either a replicate screen (**b**) or bulk RNA-Seq for *NRF1* exon-7 (**d**) and *TAF5* exon-8 (**f**) perturbations. Each point represents a gene; Two-sided Pearson correlation coefficients with corresponding significance values are indicated.

were analyzed as described above by pseudo-aggregating cells harboring the same hgRNA, followed by differential gene expression analysis. In total, we profiled 692 cells targeting *NRF1* exon-7 and 4487 cells carrying intergenic hgRNAs. This replicate experiment shows a strong overlap in differentially expressed genes with the original screen ($p = 5.54$e-90, odds ratio = 30.9, Fisher's exact test), and a near-perfect correlation in log2 fold-change values ($R = 0.96$; Fig. 6a, b), demonstrating high robustness and reproducibility.

We next benchmarked scCHyMErA-Seq against orthogonal perturbation approaches. Comparison with genome-scale Perturb-seq datasets for genes targeted by both methods[28] reveals a significant overlap in differentially expressed genes ($p < 2.2$e-16, odds ratio = 5.28, Fisher's exact test; Supplementary Fig. 20a), and strong concordance in transcriptional responses, including a high correlation in log2 fold-change values across shared targets ($R = 0.76$; Supplementary Fig. 20b, c). Notably, scCHyMErA-Seq detects a larger number of differentially expressed genes, likely reflecting more efficient and complete gene inactivation through dual Cas9/Cas12a targeting, despite the shorter experimental duration (4 days post-transduction versus eight days for Perturb-seq[28]).

Finally, we benchmarked exon-level perturbations against orthogonal strategies used to generate the ΔE7-NRF1 isoform, such as isoform rescue and base-editing approaches coupled with bulk RNA-Seq. Despite differences in the cell lines used between the bulk RNA-Seq (i.e., HEK293T) and the scCHyMErA-Seq (i.e., HAP1) experiments, we observe a strong overlap in differentially expressed genes ($p = 1.27$e-97, odds ratio = 40.1, Fisher's exact test; Fig. 6c) and a high correlation in log2 fold-change values ($R = 0.86$; Fig. 6d). Similar concordance is also observed for *TAF5* exon-8, where focused RNA-Seq data from isoform rescue experiments[17] significantly overlap with scCHyMErA-Seq results and show a strong positive log2 fold-change correlation ($p = 7.65$e-45, odds ratio = 5.4, Fisher's exact test; $R = 0.86$; Fig. 6e, f).

Collectively, these data underscore the sensitivity, specificity, and reproducibility of scCHyMErA-Seq for high-throughput exon deletion coupled with transcriptome profiling. Overall, scCHyMErA-Seq enables scalable mapping of transcriptional phenotypes arising from combinatorial genetic perturbations, including exon deletions, establishing it as a robust and versatile alternative to conventional gene-level knockout and perturbation screening approaches.

## Discussion

We have developed scCHyMErA-Seq, a combinatorial CRISPR perturbation platform that leverages highly efficient Cas9 and Cas12a editing to generate precise gene segment deletions while capturing transcriptional phenotypes at single-cell resolution. By directly linking targeted exon perturbations to downstream transcriptional consequences, scCHyMErA-Seq expands the scope of functional genomics to systematically interrogate exon-level gene regulation.

A key technical advance of scCHyMErA-Seq is the modification of the Cas12a guide scaffold to enable direct capture of Cas12a gRNAs.

These changes stabilize guide RNAs and improve expression by preventing spurious RNA polymerase III termination at T-rich sequences within the Cas12a DR. Similar optimization strategies have improved Cas9 tracrRNA performance[64] and motivated alternative approaches such as riboswitch incorporation[31] or recruiting of RNA-binding proteins to stabilize polyuridylated Pol III transcripts[33]. Our findings underscore that, in addition to transcript stability, T-rich stretches can substantially limit guide expression, an effect that becomes particularly pronounced in longer guide cassettes used for multiplexed Cas12a or CHyMErA-based systems. This principle may extend to other CRISPR platforms, including RNA-targeting systems such as Cas13[65], which hold considerable promise for systematic exon perturbation at the RNA level[66–71].

Applying scCHyMErA-Seq, we systematically interrogated 161 genes and 224 exons, identifying numerous exons with strong and specific transcriptional functions. Perturbations cluster according to shared pathways and protein complexes, revealing coordinated exon- and gene-level regulatory programs. For example, deletion of *TAF5* exon-8 or *LSM11* exon-3 disrupt histone 3′ end processing and leads to the accumulation of aberrantly polyadenylated histone mRNAs, illustrating how individual exons can exert discrete and mechanistically interpretable regulatory effects.

Our previous large-scale exon-perturbation screens also revealed a significant enrichment of fitness-promoting exons within genes involved in RNA processing and gene expression[17]. Coupling these perturbations with scRNA-Seq readouts enables systematic functional annotation of regulatory exons and provides a powerful framework for dissecting transcription factor regulation. This is particularly relevant given that transcription factors and specific splice isoforms are frequently implicated in cancer and are emerging therapeutic targets[72,73]. To our knowledge, the transcriptomic pseudobulk analyses presented here—covering more than 200 individual exon deletions—represent the most comprehensive resource to date for assessing gene expression changes resulting from exon skipping. Validation using orthogonal perturbation strategies demonstrates consistent strong concordance between scCHyMErA-Seq and low-throughput bulk RNA-seq approaches, including isoform rescue and base editing. These experiments further establish that *NRF1* exon-7 inclusion is required for efficient NRF1 recruitment to target gene promoters, underscoring the functional importance of alternative splicing in transcriptional regulation.

Alternative splicing is dynamically regulated across the cell cycle and influences numerous genes essential for cell cycle control[46]. These periodic splicing programs are coordinated in part by the SR protein kinase CLK1 and modulated by kinases such as Aurora-A[46,74,75]. Our study identifies dozens of exons whose perturbation alters cell cycle dynamics, providing causal evidence linking alternative splicing to cell state transitions. Extending scCHyMErA-Seq to developmental and tissue-specific splicing programs, therefore, holds great potential for uncovering how RNA processing shapes cell fate and organismal development, as supported by recent work identifying functionally critical microexons in neurodevelopment and autism[76].

Finally, the modularity and scalability of CHyMErA make it a versatile platform for combinatorial perturbation studies, including genetic interaction mapping and targeted interrogation of specific genomic elements. Integration with long-read RNA-sequencing technologies may further expand its utility beyond gene expression, enabling systematic analysis of RNA splicing, post-transcriptional[77], and epitranscriptomic processes.

## Methods

### Cell culture
HEK293 cells and HAP1 cells expressing *Sp*Cas9 and opCas12a[17] were cultured in Dulbecco's modified Eagle's medium (DMEM) with high glucose and pyruvate (Gibco #11995073), supplemented with 10% heat-inactivated fetal bovine serum (HI-FBS) (Gibco #16140071) and 1% penicillin-streptomycin (Gibco #15140122). Cells were cultured at 37 °C in a humidified atmosphere with 5% $CO_2$ in a HERA VIOS incubator (Thermo Fisher Scientific). Cells were passaged every 3rd day by collecting cells with 0.25% trypsin-EDTA (Gibco #25200056), counting and reseeding them in culture media on TC plastic. Cell confluence and viability was monitored using the EVOS M5000 Imaging System (Invitrogen) and the Vi-Cell BLU Cell Viability Analyzer (Beckman Coulter), respectively.

### Lentivirus production
Lentivirus was generated as described previously[14]. In brief, for library virus production, 8 million HEK293T cells were seeded in 15-cm plates. The next day, cells were transfected with 8 µg of lentiviral pLCHKO library, 4.8 µg psPAX2 virus packaging, and 3.2 µg pMD2.G virus envelope plasmids using 48 µL of X-tremeGENE 9 DNA Transfection Reagent (Sigma-Aldrich #6365809001), and 800 µL of Opti-MEM medium (Gibco #31985062). 20 h later, the medium was replaced with serum-free virus harvest media (DMEM, 1.1 g/100 mL BSA, 1% penicillin/streptomycin). The virus-containing supernatant was collected 48 post-transfection, centrifuged at 475 × g for 5 min at 4 °C, aliquoted, and stored at −80 °C. Lentiviral titers were determined by transducing cells with a titration of library virus along with polybrene (8 µg/mL) (Sigma-Aldrich #H9268). Twenty-four hours after transduction, cells were selected with 2 µg/mL puromycin for 48 h. Cell counts were subsequently compared to virus-transduced, but non-selected cells, to estimate the multiplicity of infection (MOI) of a given viral volume.

For focused hgRNA production, 750k HEK293T cells were seeded in six-well plates and transfected with 1 µg pLCHKO, 600 ng psPAX2 and 400 ng pMD2.G vectors using 6 µL of X-tremeGENE and 100 µL Opti-MEM.

### Cloning of focused hgRNAs into pLCHKO vectors
Hybrid gRNAs for focused experiments were cloned into pLCHKO vectors as previously described[14]. In brief, hgRNAs containing the required scaffold (e.g., tracrRNA, CS, and DR) were synthesized as double-stranded DNA fragments (TWIST Biosciences) and cloned into the appropriate pLCHKO vector via GoldenGate cloning using BveI and T4 DNA ligase. The fragments used are listed in Supplementary Data 8.

### Cloning of pLCHKOv4 vectors
To facilitate hgRNA cloning for single-cell RNA-Seq applications, a modified version of the pLCHKOv3 vector backbone was cloned, containing CS2 and tevopreQ1 sequences. pLCHKOv3 (Addgene #209025) was digested with AarI (Thermo Fisher Scientific #ER1581) and BspI407 (Thermo Fisher Scientific #ER0931), purified on a PCR purification column, and a dsDNA fragment (Supplementary Data 8, TWIST Biosciences) containing the CS2-tevopreQ1 sequence was cloned into the digested vector using In-Fusion Snap Assembly Master Mix (Takara Bio #638948) creating pLCHKOv4 (Addgene # 237451). A similar cloning strategy has been used for other tested hgRNA variations (Supplementary Data 8).

### Cloning of Topo vector with tracrRNA-CS1-DRv10
The CS1 and DRv10 sequences were introduced into Topo_SpCas9.-tracr_AsCas12a.DR (Addgene #155050) using homology-based cloning. First, the Topo vector was digested with Esp3I and gel extracted to remove the old tracrRNA-DR segment. Then the new *Sp*Cas9.tracr-CS1-*As*Cas12a.DRv10 fragment was introduced via a dsDNA fragment (Supplementary Data 8) using NEBuilder HiFi DNA Assembly Master Mix (New England Biolabs #E2621L), creating Topo_SpCas9.tracr-CS1_AsCas12a.DRv10 (Addgene # 237553).

## Generation of base editor vectors

To construct transfection-based all-in-one *Sp*Cas9 base editor plasmids, we used pX459v2 (Addgene #62988) as the backbone vector. The ABE8e and evoCDA1 base editors, along with nickase *Sp*Cas9 (n*Sp*Cas9), were obtained from Addgene plasmids #209044 and #209042, respectively. The wild-type Cas9 sequence in pX459v2 was replaced with the corresponding base editor and n*Sp*Cas9 sequences using standard cloning techniques. This generated the new base editor plasmids pX461-ABE8e-n*Sp*Cas9 (Addgene #237460) and pX461-evoCDA1-n*Sp*Cas9 (Addgene #237461). To generate the pX462-ABE8e-n*Sa*Cas9 base editor backbone plasmid (Addgene #237462), we first replaced the *Sp*Cas9 tracrRNA sequence in pX459v2 (Addgene #62988) with the *Sa*Cas9 tracrRNA sequence. Subsequently, ABE8e and n*Sa*Cas9 sequences were PCR-amplified and cloned into the modified backbone using NEBuilder HiFi DNA Assembly Master Mix (New England Biolabs, #E2621L). The sgRNA sequences used for targeting NRF1 exon-7 and intergenic control regions are provided in Supplementary Data 8.

## Cloning CHyMErA exon deletion lentiviral library

The scCHyMErA-Seq exon deletion library was cloned into a lentiviral backbone using a two-step cloning protocol as described previously[14], with some library-specific modifications described below. First, a 120-nt hgRNA library oligo pool containing 1066 Cas9 and Cas12a spacer sequences flanked by BveI restriction sites and intervened by a stuffer sequence with Esp3I restriction sites was synthesized (TWIST Biosciences). The oligo pool was amplified with KAPA HiFi HotStart DNA Polymerase (Roche #KK2601) using customized primers (Supplementary Data 8) in an Applied Biosystems Veriti 96-well thermal cycler using the following cycling conditions: step 1: 95 °C for 3 min; step 2: 98 °C for 15 s, 65 °C for 15 s, 65 °C for 45 s; step 2 was repeated for a total of seven cycles. The PCR product was purified on a PCR purification column (Thermo Fisher Scientific #K0701), and the amplicon size was confirmed by agarose gel electrophoresis. Next, the pLCHKOv4 vector (Addgene #237451) harboring the 10x Genomics capture sequence 2 (CS2) and tevopreQ1 riboswitch was digested with BveI (Thermo Fisher Scientific #FD1744) and the vector backbone was gel-purified. The amplified library oligos were digested with BveI and ligated into pLCHKOv4 with T4 DNA ligase (New England Biolabs #M0202) in a GoldenGate reaction (step 1: 37 °C for 30 min; 22 °C for 60 min, step 2: 37 °C for 30 min, 22 °C for 45 min; step 2 was repeated for 12 cycles; step 3: 37 °C for 15 min; step 4: 65 °C for 20 min) using a 1:5 vector:insert molar ratio. The GoldenGate reaction was purified by ethanol precipitation, and the library was transformed into Endura competent cells (LGC Biosearch Technologies #60242-2) by electroporation (1 mm cuvette, 25 μF, 200 Ω, 1600 V), and plated on 15-cm 100 μg/mL ampicillin Luria–Bertani (LB) agar plates, resulting in a library coverage of >30,000-fold. The bacterial colonies were collected, and the ligation 1 plasmid library was purified using Qiagen Endotoxin-free Maxi Prep (Qiagen #12362).

In a second step, the *Sp*Cas9 tracrRNA with CS1 followed by a *As*Cas12a direct repeat v10 (DR$_{v10}$) was cloned into the intermediate library to complete the single-cell hgRNA expression cassette. The ligation 1 plasmid was digested with Esp3I (Thermo Fisher Scientific #FD0454), dephosphorylated using FastAP (Thermo Fisher Scientific #EF0651), concentrated on a PCR purification column and subsequently purified by gel extraction. The tracrRNA-CS1-DR fragment was cloned into the library in a GoldenGate reaction by digesting a TOPO vector (Addgene #237553), providing the hgRNA scaffold flanked by Esp3I sites and ligating it into the predigested ligation 1 backbone using the same protocol as used in the first cloning step. The GoldenGate reaction was purified by ethanol precipitation, transformed into Endura competent cells and plated on ampicillin LB-agar plates, resulting in a 70,000 library coverage. Colonies were harvested, and

the final plasmid library was extracted using a maxi prep kit. The library has been deposited to Addgene (#237473).

## Single-cell RNA-sequencing experiments

HAP1 cells stably expressing *Sp*Cas9 and opCas12a were used for scCHyMErA-Seq experiments[17]. The HAP1 cell line had undergone diploidization prior to screening. Three million HAP1 cells were seeded in eight 15-cm plates and transduced with the lentiviral exon deletion library or focused hgRNA constructs at a low MOI of 0.1. For the exon deletion screen, a total of 2.4 million cells were transduced, reaching a 2400x library coverage. Twenty-four hours post-transduction, cells were selected with 2 μg/mL puromycin for ~40 h. Transduced cells were collected and seeded at 1 million cells in six-well plates (T0). Twenty-four hours later (T1), cells were collected, counted, and a single-cell suspension was prepared following the wash steps described in the 10x Genomics Demonstrated Protocol "Single Cell Suspensions from Cultured Cell Lines for Single Cell RNA Sequencing" (CG00054 RevB).

For single cell capture and sequencing library construction, the protocol described in the 10x Genomics User Guide "Chromium Next GEM Single Cell 3 Reagents Kits v3.1 (Dual Index)" (CG000316 RevD) was followed. GEM generation and barcoding was performed using a 10x Genomics Chromium X device using eight GEM replicates for the exon deletion screen. Gene expression libraries were constructed as recommended by 10x Genomics. The CRISPR screening library construction was performed as described in the user guide, whereby 11 cycles with 5 μL input were performed for the feature PCR, and nine cycles with 10 μL input were applied for the sample index PCR.

To enhance Cas12a capture, a customized Cas12a screening library protocol was applied. First, a Cas12a custom feature PCR was performed using 20 μL CRISPR cDNA cleanup product (step 4.1 m, 10x Genomics), 2.5 μL of 10 μM customized primers (Supplementary Data 8), and 50 μL of 2x Amp Mix (10x Genomics) in a total volume of 100 μL (step 1: 98 °C for 45 s; step 2: 98 °C for 20 ss, 54 °C for 5 s, 72 °C for 5 s; step to was repeated for 16 cycles; step 3: 72 °C for 60 s). The PCR product was cleaned up using the standard 10x Genomics post-feature PCR cleanup protocol (step 4.3) using SPRIselect beads (Beckman Coulter #B23318). Next, a sample index PCR was performed for ten cycles with 10 μL of input using 10x Genomics reagents following the standard 10x Genomics protocol (step 4.4). Finally, post sample index PCR double-sided size selection was performed as described (step 4.5), and 1 μL of samples were analyzed on TapeStation D1000 Screen Tape (Agilent #5067-5582).

For sequencing, gene expression, CRISPR (Cas9 and Cas12a gRNAs) and Cas12a sequencing libraries were pooled at a 4:1:1 molar ratio. Samples were sequenced on a NextSeq or NovaSeq platform using a 100-cycle kit and the standard run recipe for 10x Genomics libraries: Read 1: 28 bp, Index i7: 10 bp, Index i5: 10 bp, Read 2: 90 bp.

## Downstream processing of scCHyMErA-seq data

The alignment and processing of raw 10x Genomics data from eight replicates was performed using Cell Ranger software (10x Genomics, v8.0.1). Sequencing reads from the gene expression (GEX) library were aligned to GRCh38 (refdata-gex-GRCh38-2024-A), and gRNA (CRISPR and Cas12a-specific) library reads were aligned to the feature reference file specified with --feature-ref option using the "cellranger count" function. Cell Ranger's aggr function was used for combining the output from the eight replicates. The generated feature-barcode matrix was then loaded into Scanpy (v1.10.4)[78] for downstream analyses. Quality control was done using the standard QC matrix, UMI count and number of expressed genes and percentage mitochondrial count, computed from scanpy.pp.calculate_qc_metrics function. The cells exhibiting less than 200 expressed genes and genes expressed in less than three cells were removed from the matrix. Any cell expressing

non-targeting gRNAs were also excluded. We selected cells that had only one hgRNA targeting a specific feature.

The cells expressing hgRNAs for exon perturbation and gene knockout were separated for further analysis. Gene expression counts were normalized to 10,000 reads per cell (scanpy.pp.normalize_total) and log transformed (scanpy.pp.log1p). Highly variable genes were selected (scanpy.pp.highly_variable_genes with min_mean = 0.0125, max_mean = 3, min_disp = 0.5). To mitigate technical biases, unwanted variation from total counts and mitochondrial transcript percentage was regressed out (scanpy.pp.regress_out). Counts were then scaled to unit variance and zero mean, with an upper limit of 10 (scanpy.pp.scale(max_value = 10). Next, we applied the mixscape pipeline implemented in pertpy[38] version 0.9.4 (ms = pertpy.tl.Mixscape()) to determine the perturbed cells. To correct for confounding effects, local perturbation signatures were computed and regressed out using "ms.perturbation_signature". Dimensionality reduction was performed using PCA (scanpy.pp.pca), followed by neighborhood clustering (scanpy.pp.neighbors) and UMAP embedding (scanpy.tl.umap). Cells with detectable perturbations were identified and selected with mixscape (ms.mixscape with control = "intergenic"). Clustering was carried out using the Leiden algorithm (sc.tl.leiden with resolution = 0.15, flavor = "igraph", n_iterations = 2, and directed = False). The proportion of differential perturbation tags in each Leiden cluster was assessed using Fisher's exact test. Finally, linear discriminant analysis (LDA) was applied (ms.lda), and UMAP was calculated (ms.plot_lda) to visualize the maximally separated clusters for each perturbation.

To gain confidence in the consistency of guide replicates targeting a particular exon or gene, we first aggregated read counts for each guide by implementing decoupler.get_pseudobulk. Then, differentially expressed genes were determined against intergenic control by applying the glmQLFTest function in the edgeR package[79]. The log2 fold change of highly variable genes were used to visualize the correlation across the guides. For exon perturbation, with three guides, we first calculated all pairwise correlations and then take the mean of the correlations.

### Pseudobulk analysis
To determine differentially expressed genes (DEGs) for each perturbation/knockout class in comparison to the intergenic control, we generated a pseudobulk count matrix. The decoupleR package[80] was used to create the pseudobulk profiles by aggregating the read counts across each class (decoupler.get_pseudobulk). Differential expression analysis was done using a Python implementation of the DESeq2 pipeline (PyDESeq2)[81]. Genes with adjusted $p$ value <0.05 and |log2 fold-change| >0.5 were defined as significant. In Fig. 5d, we showed the Pearson correlation coefficients between log2 fold-change of the union of DEGs in exon perturbation and knockout cases. In addition, to identify exons with distinct phenotypic effects from the gene-level impact, we used pseudobulk profiles of each exon and their corresponding knockout to calculate DEG by PyDESeq2.

### Gene ontology analysis
A Python script was used to generate filtered gene lists, selecting genes with an adjusted $p$ value <0.05 and a log2 fold-change below −0.5 or above 0.5. A custom statistical domain scope was defined using a background gene list comprising all genes with an adjusted $p$ value <0.05. Functional profiling was then performed using an R script, leveraging g:Profiler's g:GOSt tool[82]. Significant results were collected for all perturbations. The scripts are available on a dedicated GitHub repository (see Code availability). For heatmap visualization, GO:BP, GO:MF, and KEGG terms were included if they contained 250 or fewer genes and after collapsing similar terms using Revigo[83].

Functional enrichment analysis of differentially expressed genes (DEGs) resulting from exon perturbations relative to intergenic controls was performed using g:Profiler (g:GOSt). For SKP2 exon-6, g:GOSt was applied to DEGs identified relative to the SKP2 knockout in two gene sets: 119 DEGs across all cells and 51 DEGs specific to G1-phase cells; no enriched terms were identified for the G1-phase-specific set. As a background gene set, expressed genes were defined from the Cell Ranger filtered feature–barcode matrix by excluding cells with fewer than 200 detected genes and genes detected in fewer than three cells. Gene-level expression was calculated by summing UMI counts across all quality-controlled cells, and genes with at least one UMI per 100 profiled cells (UMI ≥2122) were considered expressed. In total, 17,288 coding and non-coding genes met this criterion and were used as the background set using the "custom over all known genes" option in g:Profiler.

### Hierarchical clustering across CRISPR perturbations
Differentially expressed genes (DEGs) were identified based on a log2 fold-change >0.5 and an adjusted $p$ value ($P$adj) <0.05. Genes that were common across at least 30 perturbations (out of 351 gene-level knockouts and exon deletions combined) were selected. Hierarchical clustering was performed using Pearson correlation of log2 fold-change values across all or only gene knockout perturbations to generate clustermaps. Functional enrichments of subcluster-specific genes from the clustermaps were performed with clusterProfiler[84] ORA (over-representation analysis) and g:Profiler[82,85] GSEA (gene set enrichment analysis) methods with MSigDB gene sets obtained using msigdbr. $P$ values were calculated using hypergeometric distribution statistics, which is equivalent to a one-sided Fisher's exact test. Multiple hypothesis test corrections were performed using the Benjamini–Hochberg (BH) method, and gene sets, or pathways with BH-adjusted $p$ value <0.05 were considered significant. The top hits were chosen for subcluster annotation included in the figures.

### Structural predictions of full-length and exon-deleted isoforms
To assess the structural impact of alternative isoforms, protein structures of full-length and exon-deleted variants were predicted using AlphaFold2[86,87] (v2.3.1). Amino acid sequences for full-length transcripts were obtained from the UCSC Genome Browser for NRF1 (ENST00000393232.6), SKP2 (ENST00000274255.11), SMG5 (ENST00000361813.5), NUBP1 (ENST00000283027.10), TFAM (ENST00000487519.6), and ESPL1 (ENST00000920820.1). Annotated exon-deleted isoforms were used for SMG5 Δexon-5 (ENST00000858608.1), NUBP1 Δexon-4 (ENST00000960379.1), and TFAM Δexon-5 (ENST00000935270.1). For NRF1, SKP2, and ESPL1, exon-deletion isoforms were generated in silico by removing exon-7, exon-6, or exon-8, respectively, from the corresponding full-length genomic DNA sequences obtained from UCSC. Open reading frames were identified using ORFinder (NCBI), and the resulting predicted protein sequences were used for structure prediction. AlphaFold2 predictions were performed using default parameters with --model_preset=monomer. The --max_template_date was set to 2025-12-17 for all proteins except NRF1, for which it was set to 2024-06-13. For each protein, the highest-confidence structural model (ranked_0.pdb) was selected for downstream analysis and visualization using UCSF ChimeraX[88].

### Targeted exon translational output assessment
The genomic coordinates of exons targeted for deletion were used to identify annotated human gene transcripts that either include or skip the corresponding exon, based on the GENCODE v49 GTF annotation (Supplementary Data 7). APPRIS annotations for "PRINCIPAL" and "ALTERNATIVE" isoforms were retrieved using the biomaRt tool in R. Protein-coding sequences and corresponding peptide identifiers for each transcript were extracted using the PyEnsembl tool in Python, leveraging the GENCODE v49 transcript sequence FASTA and protein translation FASTA files.

For exon-skipping isoforms, the coding sequence and reading frame were re-evaluated to determine the predicted impact of CRISPR-mediated perturbation on protein translation. In the majority of cases, exon skipping preserved the coding frame, resulting in proteins lacking only the amino acids encoded by the targeted exon. For mutually exclusive exon (MXE) pairs in the *PKM* and *FGFR2* genes, skipping of the targeted exon was determined to lead to inclusion of the corresponding alternative MXE, thereby maintaining the coding frame.

Strict exon-skipping transcripts (Supplementary Data 7) were selected when transcripts exhibited precise alterations in their coding sequences attributable to the targeted exon, namely deletion of the targeted exon for exon-skipping events or replacement by the alternative MXE in the case of targeted MXEs, with all other coding regions remaining unchanged. The longest exon-skipping mRNA isoform was selected based solely on exclusion of the targeted exon, irrespective of additional differences elsewhere in the transcript.

### Cell cycle analysis of scCHyMErA-seq data

To assess the effect of exon perturbation on cell cycle progression, we assigned a cell cycle phase to each cell using scanpy.tl.score_genes_cell_cycle[89]. Cell cycle marker genes were obtained from https://github.com/scverse/scanpy_usage/tree/master/180209_cell_cycle/data[49]. For each exon, the total number of cells in G1, S, and G2/M phases was calculated from Mixscape-passed cells. Significant cell cycle phases were then tested relative to intergenic controls using one-sided Fisher's exact and Mann–Whitney tests (exact=FALSE, correct=FALSE). *P* values were adjusted using the Benjamini–Hochberg method, with cutoffs of 0.01 for Fisher's test and 0.05 for the Mann–Whitney test. A cell cycle phase was considered significant for an exon only if it passed both tests.

A similar analysis was performed for gene knockouts; however, due to the lack of significant results from the Mann–Whitney test, significance was determined based on Fisher's test with an adjusted *p* value <0.01. In cases where two phases were significant, the phase with the lowest adjusted *p* value was selected.

### Functional enrichment (GO) analysis for *SKP2* exon-6 regulated genes

Functional annotation of DEGs, identified for SKP2 exon-6 relative to the SKP2 knockout, was performed using g:Profiler web interface (g:GOSt function). We ran g:Profiler with two DEG sets: 119 DEGs identified across all cells and 51 DEGs specific to G1-phase cells. For the background set, we used all genes identified as expressed in the exon-versus-knockout comparisons and selected the "custom over all known genes" option.

### Comparison of scCHyMErA-Seq and Perturb-seq performance

We obtained raw single-cell expression data for K562 genome-scale Perturb-seq sampled at day 8 post-transduction, K562_gwps_raw_singlecell_01.h5ad from[28]. The comparison was restricted to the cells that were targeted by the same genes as in the scCHyMErA-Seq knockout experiment. As previously described, we loaded the.h5ad file into Scanpy, removed cells with <200 expressed genes and genes expressed in <3 cells from the matrix, then used dcoupler.get_pseudobulk to create a pseudobulk profile for each guide. We then created pseudobulk for the scCHyMErA-Seq knockout data after merging the read counts of guide replicates. Considering the absence of replicates, we applied glmQLFTest in edgeR to determine DEGs against provided non-targeting control in Perturb-seq experiment and intergenic for scCHyMErA-Seq. Finally, we plotted a heatmap showing the Spearman's correlation of log2 fold change for all the expressed genes common in both experiments. DEGs were selected from genes with adjusted *p* value <0.05 and log2 fold change >0.5.

### Northern blotting

Total RNA was extracted with TRIzol reagent (Sigma-Aldrich #T3934) according to the manufacturer's protocol. For each sample, 5 µg of RNA was denatured in loading dye (Thermo Fisher Scientific #LC6876) at 95 °C for 5 min, then chilled on ice for 5 min to minimize secondary structures. RNA was resolved on a 10% TBE-Urea gel (Thermo Fisher Scientific #EC68752BOX) at 200 V for 1.5 h. Following electrophoresis, RNA was transferred to a Hybond-N+ membrane (Amersham #NV0796) using a semi-dry transfer system (Bio-Rad #1703940) at 300 mA for 1.5 h. Ethidium bromide staining (0.2%) was performed to confirm equal loading across samples. Membranes were UV crosslinked (254 nm, 120,000 µJ/cm², twice; VWR #89131-484) and pre-hybridized in ULTRAhyb-Oligo buffer (Thermo Fisher Scientific #AM8663) for 45 min at 42 °C with gentle rotation. Hybridization was performed overnight at 42 °C with 4 µL of 10 µM biotin-labeled oligonucleotide probes (Supplementary Data 8). Detection was carried out using the North2South Chemiluminescent Hybridization and Detection Kit (Thermo Fisher Scientific #17097), and signals were visualized on an iBright CL1500 Imaging System (Invitrogen).

### Western blotting

Cells were washed once with DPBS (Thermo Fisher Scientific #14190250) and lysed on ice for 10 min in F buffer (10 mM Tris-HCl, pH 7.05; 50 mM NaCl; 30 mM sodium pyrophosphate; 50 mM NaF; 5 µM ZnCl$_2$; 10% glycerol; 0.5% Triton X-100) supplemented with protease inhibitors (Roche #11836170001). Lysates were clarified by centrifugation at 18,500 × *g* for 10 min at 4 °C. Protein concentrations were determined using Bradford reagent (Bio-Rad #5000006), and equal amounts of protein (10–30 µg) were separated on 4–12% Bis-Tris SDS–PAGE gels (Life Technologies #NP0323BOX). Proteins were transferred to PVDF membranes (Immobilon-P, Millipore #IPVH00010) using a Mini Blot Module (Life Technologies) at 22 V for 60 min. Membranes were blocked with 5% non-fat milk for 1 h at room temperature and incubated overnight at 4 °C with primary antibodies against NRF1 (1:2000; Proteintech #12482-1-AP), TAF5 (1:2000; Thermo Fisher Scientific #MA3-076), SRSF3 (1:1000; MBL Life Science #RN080PW), FLAG (1:2500; Sigma-Aldrich #F3165), GAPDH (1:2500; Proteintech #10494-1-AP), and β-tubulin (1:2500; Proteintech #10094-1-AP). After washing three times, membranes were incubated with HRP-conjugated secondary antibodies (anti-rabbit, Cell Signaling Technology #7074; anti-mouse, Cell Signaling Technology #7076) diluted 1:5000 for 1 h at room temperature. Chemiluminescent signals were detected using SuperSignal West Pico PLUS substrate (Thermo Fisher Scientific #34580) and imaged with the iBright CL1500 Imaging System (Invitrogen).

### Immunofluorescence analysis

Cells cultured on coverslips were fixed with 4% paraformaldehyde (Thermo Fisher Scientific #28908) for 30 min at room temperature. After fixation, cells were permeabilized in 0.2% Triton X-100 (Sigma-Aldrich #T8787) for 10 min, followed by blocking in 5% BSA (Sigma-Aldrich #A9647) for 1 h. Primary antibodies against NRF1 (1:500; Proteintech #12482-1-AP) and FLAG (1:500; Sigma-Aldrich #F3165) were applied for 1 h at room temperature in DPBS containing 5% BSA. After washing, cells were incubated with secondary antibodies: Alexa Fluor 594-conjugated goat anti-rabbit (1:1000; Thermo Fisher Scientific #A32740) and Alexa Fluor 488-conjugated goat anti-mouse (1:1000; Thermo Fisher Scientific #A32723) for 1 h at room temperature. Nuclei were counterstained with Hoechst 33342 (1:10,000; Thermo Fisher Scientific #62249), and slides were mounted using Fluoromount-G™ (Thermo Fisher Scientific #00-4958-02). Imaging was performed with a Leica TCS SP8 confocal microscope using a 63x oil-immersion objective. Images were processed with Fiji (ImageJ).

## NRF1 isoform rescue experiments

Four siRNAs targeting NRF1 (Dharmacon #LQ-017924-00-0005) were screened for knockdown efficiency by qRT-PCR and western blot. The most effective siRNA (Dharmacon #J-017924-08-0005; sequence: GCUAUUGUCCUCUGUAUCU) was used for subsequent experiments. HEK293 Flp-In cells were transfected with NRF1 siRNA using Lipofectamine RNAiMAX (Thermo Fisher Scientific #13778150). After 6 hours, doxycycline was added ($10^{-4}$ μg/mL for NRF1-FL and NRF1-ΔEx7; Sigma, #D9891) to induce expression of siRNA-resistant NRF1 isoforms: NRF1-FL (Addgene #237453) or NRF1-ΔEx7 (Addgene #237454). Cells were harvested 48 h post-induction for chromatin, RNA, or protein isolation.

## RT-PCR and qRT-PCR assays

Total RNA was isolated from frozen cell pellets using the RNeasy Plus Universal Mini Kit (Qiagen #74136), following the manufacturer's instructions. For splicing RT-PCR assays, 40 ng of RNA was subjected to reverse transcription and PCR amplification using the Qiagen One-Step RT-PCR Kit (Qiagen #210215). Primer sequences for NRF1 and other cell cycle-related splicing events are listed in Supplementary Data 8.

For most qPCR assays, 40 ng of total RNA was analyzed using the SensiFAST SYBR No-ROX One-Step Kit (Thomas Scientific #C755H72), following the manufacturer's instructions. For histone mRNA quantification, cDNA was synthesized from 1 μg of RNA and Oligo(dT) Primer using the Maxima H Minus First Strand cDNA Synthesis Kit (Thermo Fisher Scientific #K1651), diluted 1:25, and 2 μL (equivalent to 4 ng RNA) was used per reaction with the SensiFAST SYBR No-ROX Kit (Thomas Scientific #C755J02). All reactions were performed in triplicate. Primer sequences are listed in Supplementary Data 8.

## Exon deletion activity assays

HAP1 cells stably expressing *Sp*Cas9 or opCas12a variants were transduced with three independent *CD46* exon-3-targeting hgRNAs, as outlined in Fig. 1c. Seventy-two hours post-selection, cells were washed with PBS and detached using Accutase (Sigma-Aldrich #A6964) for 3 min at room temperature. The cell suspension was quenched with culture media and gently pipetted to achieve a single-cell suspension. Approximately 0.5 million cells per sample were transferred to cell strainer-cap flow tubes (Falcon #352235). Cells were washed with ice-cold flow buffer (PBS + 2% FBS), centrifuged for 3 min at 170 × *g* at 4 °C, and stained with CD46-BV421 antibody (1:100; BD Biosciences #743776) on ice for 15 min. After washing, cells were incubated with Zombie NIR viability dye (1:1000; BioLegend #423106) for 15 min on ice, followed by another wash. Cells were then fixed with 1% paraformaldehyde (Thermo Fisher Scientific #28908) for 15 min and washed twice with chilled flow buffer. Samples were analyzed on a BD LSRFortessa (BD Biosciences) using a 407 nm violet laser (450/50 filter) for BV421 and a 633 nm red laser (716/40 filter) for viability. Data were processed in FlowJo (version 10.10), with gating applied to exclude doublets and dead cells (Supplementary Fig. 21a). CD46 knockout gates were defined using unstained controls.

For agarose PCR-based assays, selected cells were lifted and reseeded in media containing 1 μg/mL puromycin for 72 h. Genomic DNA was then extracted using the GeneJet Genomic DNA Purification Kit (Thermo Fisher Scientific #K0722). Deletion of *Ptbp1* exon-8 and *Parp6* exon-20 was evaluated by PCR with PrimeSTAR Max DNA Polymerase (Takara Bio #R045B) and primers flanking the target region (Supplementary Data 8). PCR conditions were: 95 °C for 1 min; followed by 36 cycles of 98 °C for 20 s, 60 °C (*Parp6*) or 68 °C (*Ptbp1*) for 15 s, and 72 °C for 2 min; with a final extension at 72 °C for 2 min. Amplicons were separated on a 1.5% agarose gel and visualized by gel electrophoresis. Exon deletion frequency was quantified using ImageJ, measuring the intensity of the exon-excluded band relative to the total (exon-included + exon-excluded) band intensities. Values were multiplied by 100 and rounded to the nearest whole number.

## Generation of HEK293 Flp-In cell lines expressing NRF1 splice isoforms

HEK293 Flp-In cells ($1 × 10^6$) were transfected with pcDNA5-NRF1 plasmids encoding full-length (FL) or Δexon-7 (ΔEx7) NRF1 isoforms, or control vectors (EGFP), using X-tremeGENE 9 DNA Transfection Ragent according to the manufacturer's recommendations. After 4.5 h, the transfection media was replaced with fresh growth media to minimize toxicity. Forty-eight hours post-transfection, cells were selected with 200 μg/mL Hygromycin B (Thermo Fisher Scientific #10687010) in DMEM. The selection media was refreshed every 4–5 days until stable colonies expanded. Doxycycline titration was then performed to optimize expression of FLAG-tagged proteins, with final concentrations of 0.1 ng/mL for both FLAG-NRF1-FL and FLAG-NRF1-ΔEx7 constructs.

## Base-editing experiments

HEK293T cells were transfected with all-in-one base editor plasmids designed to target either an intergenic region (Addgene #237463-237465) or the splice sites of NRF1 exon-7 (Addgene #237466-237468). Twenty-four hours post-transfection, cells were subjected to puromycin selection (2 μg/mL) for 48 h. Following selection, cells were harvested for RNA and protein extraction.

## Bulk RNA sequencing and analysis

Total RNA was isolated from HEK293 cells (Flp-In stable cell lines or cells transfected with base editor constructs) using the RNeasy Plus Mini Kit (Qiagen #74136). Libraries were prepared with the Illumina Stranded mRNA Prep Kit (#15031047) and sequenced on an Illumina NovaSeq 6000 S1 flowcell (200-cycle kit). Each sample yielded an average of 93 million pass-filter reads, with over 89% of bases scoring Q30 or higher. The sequencing reads were trimmed of adapters and low-quality bases using Cutadapt (version 1.18)[90]. The trimmed reads were mapped to the human reference genome (hg38) and the annotated transcripts (GENCODE v30) using the STAR aligner (v2.7.0 f) with two-pass alignment option[91]. Differential expression analysis was conducted with DESeq2[92].

## MiniTurboID mass spectrometry analysis

The HEK293 Flp-In cells expressing miniTurbo-NRF1 isoforms (full length or Δexon-7) were processed as mentioned previously[17]. The positive interactors of NRF1 were identified by filtering against the control groups (i.e., empty vector or EGFP) with a minimum ratio of 2 and a maximum *p*-value of 0.05 (one-tailed paired *t*-test). The protein–protein interaction network was constructed using Cytoscape[93] (version 3.10.3) and the STRING[94] (plugin version 2.2.0), utilizing the integrated network query feature. A confidence score cutoff of 0.999 was applied, with no additional interactors considered. Nodes were color-coded based on the log fold-change between NRF1-FL and NRF1-ΔEx7.

## ChIP-Seq and analysis

HEK293 Flp-In cells expressing FLAG-tagged NRF1-FL, NRF1-ΔEx7, or EGFP were induced with doxycycline ($10^{-4}$ μg/mL) for 48 h. Cells were crosslinked in 1% formaldehyde (Pierce #28908) for 10 min, quenched with 125 mM glycine (Sigma #G8898) for 5 min, washed with cold PBS, and collected by scraping. Nuclei were isolated in Farnham lysis buffer (5 mM PIPES, pH 8.0, 85 mM KCl, 0.5% NP-40) with protease inhibitors, then lysed in ChIP lysis buffer (1% SDS, 10 mM EDTA, 50 mM Tris-HCl, pH 8.1). Chromatin was sheared (Diagenode Bioruptor Plus) for 15 cycles (30 sec on/off) to 200–500 bp fragments. Chromatin (50 μg per IP) was diluted in ChIP dilution buffer (0.01% SDS, 1.1% Triton X-100, 1.2 mM EDTA, 16.7 mM Tris-HCl, pH 8.1, 167 mM NaCl) and incubated

overnight at 4 °C with 6 μg anti-FLAG M2 (Sigma-Aldrich #F1804) and 40 μL Dynabeads Protein G (Thermo Fisher Scientific #10004D). Beads were washed sequentially with low salt buffer (0.1% SDS, 1% Triton X-100, 2 mM EDTA, 20 mM Tris-HCl, pH 8.1, 150 mM NaCl), high salt buffer (same with 500 mM NaCl), LiCl buffer (0.25 M LiCl, 1% NP-40, 1% sodium deoxycholate, 1 mM EDTA, 10 mM Tris-HCl, pH 8.1), and twice with TE buffer (10 mM Tris-HCl, pH 8.0, and 1 mM EDTA). Chromatin was eluted in 1% SDS, 100 mM $NaHCO_3$ at 65 °C for 30 min. Crosslinks were reversed overnight at 65 °C, followed by RNase A and Proteinase K treatment. DNA was purified using Phenol:chloroform:isoamyl alcohol.

Libraries from two biological replicates were prepared from 11.5 ng DNA (Accel-NGS 2S Plus Kit, Swift Biosciences #21096) and sequenced on an Illumina NovaSeq 6000 (paired-end, 100 bp). Reads were trimmed (Cutadapt v1.18), aligned to hg38 (Bowtie2 v2.3.4.1), and processed with Samtools. bigWig tracks were generated with deep-Tools "bamCompare" ($log_2$ ChIP/Input, 50 bp bins). Occupancy at ±5 kb from the TSS of expressed genes was analyzed with deepTools "computeMatrix" and visualized using "plotHeatmap".

### Large-scale siRNA screen for NRF1 exon-7 splicing

A high-throughput siRNA screen targeting 60 splicing factors was performed in HEK293 cells using reverse transfection with Lipofectamine RNAiMAX (Thermo Fisher Scientific #13778150). Cells were transfected with 25 nM siRNA (Dharmacon) per target and incubated for 48 h under standard culture conditions. Total RNA was then isolated, and NRF1 exon-7 splicing was assessed by RT-PCR using Qiagen OneStep RT-PCR Kit (Qiagen #210215) and the indicated primers (Supplementary Data 8). Splicing patterns were quantified from gel images using ImageJ software. The data represent three independent biological replicates.

### Cell cycle analysis by flow cytometry

HAP1 cells transduced with intergenic controls or gene and exon perturbation hgRNAs were harvested and washed once with phosphate-buffered saline (PBS). The cells were resuspended in 0.4 mL PBS, and ice-cold absolute ethanol was added dropwise while gently vortexing to ensure uniform fixation. The cell suspension was incubated at −20 °C overnight. Following fixation, the cells were centrifuged, resuspended in 5 mL PBS, and rehydrated at room temperature for 30 min. The rehydrated cells were pelleted and resuspended in propidium iodide (Thermo Fisher Scientific #P3566) staining buffer containing 30 μM propidium iodide and 200 μg/mL RNase A (Invitrogen #12091021), followed by incubation at 37 °C for 30 min in the dark. Samples were subjected to flow cytometry (BD LSRII Fortessa analyzer), and cell cycle distribution was determined using FlowJo v10 software. Gating strategy is provided in Supplementary Fig. 21b.

### Reporting summary

Further information on research design is available in the Nature Portfolio Reporting Summary linked to this article.

### Data availability

The data supporting the findings of this study are available from the corresponding authors upon request. The scCHyMErA-Seq screening data generated in this study have been deposited to Gene Expression Omnibus (GEO) under accession code GSE293013. The NRF1-related RNA-Seq data generated in this study have been deposited to GEO under accession codes GSE292986 and GSE292988. The NRF1 ChIP-Seq data generated in this study have been deposited to GEO under accession code GSE292984. The Raw data generated in this study are provided in the Source Data file and have been deposited in Mendeley Data under the https://doi.org/10.17632/rxfzrptr2d.1[95]. All generated key plasmids and libraries have been deposited to Addgene (see

Supplementary Data 8): scCHyMErA-Seq Library; Plasmids. Source data are provided with this paper.

### Code availability

The manuscript does not report the development of a new code. However, a dedicated GitHub repository has been created to document and summarize the code used for analyzing the scCHyMErA-Seq screen data: https://github.com/NCI-RBL/scCHyMErA-Seq[96].

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

## Acknowledgements

The authors thank members of the Gonatopoulos-Pournatzis and Aregger groups, as well as the RNA Biology Laboratory and the Molecular Targets Program at NCI-Frederick for constructive discussions and technical assistance. We thank Ben Blencowe for insightful discussions and for his generous collegiality regarding the timing of related work. The authors thank Nathan Wong for reviewing the single-cell analytical pipeline. We are also grateful to the CCR Sequencing Facility. The authors thank Thorkell Anderson and Gerard Duncan for their valuable assistance with the mass spectrometry experiments. This research was supported by the Center for Cancer Research, National Cancer Institute (NCI), National Institutes of Health (NIH) Intramural Research Program project numbers ZIA BC012033, ZIA BC012101, and federal funds from the NCI, NIH under contract No. HHSN261200800001E and 75N91019D00024. The contributions of the NIH authors are considered works of the United States Government. The findings and conclusions presented in this paper are those of the authors and do not necessarily reflect the views of the NIH or the US Department of Health and Human Services, nor does mention of trade names, commercial products, or organizations imply endorsement by the US Government.

## Author contributions

Conceptualization, study design, and funding acquisition: T.G.-P. and M.A.; Data generation: A.P.D., M.-S.X., T.A.O., M.T., C.H., S.L., N.P., M.A., and T.G.-P.; Formal analysis: B.K., A.P.D., W.M.G., M.-S.X., A.K.B., T.A.O., C.E.M., C.H., Y.Z., M.A., and T.G.-P.; Supervision: T.G.-P., M.A., A.P.D., W.M.G., and M.-S.X.; Writing—original draft: T.G.-P. and M.A.; Writing—review and editing: all authors.

## Funding

## Competing interests

The authors declare no competing interests.
