## [Transparent Peer Review file · Nature Communications]

Single-Cell Exon Deletion Profiling Reveals Splicing Events That Shape Gene Expression and Cell State Dynamics

Corresponding Author: Dr Thomas Gonatopoulos-Pournatzis

Version 0:

Reviewer comments:

Reviewer #1

(Remarks to the Author)

I thank the authors for this very thorough response to reviewer comments, which has satisfied all of my previous concerns.

Reviewer #2

(Remarks to the Author)

The author's responses have addressed all of my concerns. Congratulations on this nice paper.

Reviewer #3

(Remarks to the Author)

The authors have largely addressed this reviewers primary concerns. The additional validations (qPCR, western blot), analyses (exon vs. gene, cell cycle) and methodological clarifications (i.e., haploid vs. diploid) substantially improve the rigor and qualitative assessment of the scCHyMERa-Seq.

Examples this reviewer were most interested in were those in which exon deletion results in a more significant perturbation than the gene knockout alone, suggesting an isoform with an altered function from the predominantly expressed longer isoform (Figure R18). SKP2 is an intriguing example. Can the authors further comment on ESPL1 exon 8, NUP98 exon 6 and SMG5 exon 5, NUBP1 exon 4 and TFAM exon 5, in the manuscript, which have equivalent or greater perturbation impacts? Specifically, do loss of these exons result in significant impacts to protein expression, 3D protein structure (AlphaFold) or domain composition. Several of these are key mediators of tumorigenesis in which alternative splicing has not necessarily been implicated in disease pathogenesis. These vignettes will be of most interest to biologists trying to understand, prioritize and model isoform specific impacts in a high throughput manner (these are important examples for others to potentially follow up on in the future).

The reviewer thanks the authors for adding Supplemental Table 7, which details exon deletions that could impact protein domain architecture. In Supplementary Table 1 can the authors include: 1) the longest associated protein coding Ensembl isoform or transcript (where isoform is non-coding) associated with each exon deletion (inferred) and 2) inferred protein length difference from reference isoform (APPRIS principal isoform or equivalent). This is necessary to interpret exon deletion specific or non-specific impacts.

Please include a note in the methods that the HAP1 cells have undergone diploidization.

We thank the reviewers for their thoughtful feedback and constructive suggestions, which have greatly strengthened our manuscript. In this revised version, we have incorporated new experiments and analyses that comprehensively address all comments within the scope of our study. Specifically, we expanded our single-cell RNA-seq analyses to identify exons that influence cell cycle distribution and validated these findings through targeted flow cytometry assays. We further assessed the performance of individual guides, observing highly concordant transcriptional effects and strong correlations among hgRNAs targeting the same exon or gene. We also confirmed the robustness of exon skipping through extensive reverse transcriptase (RT)-PCR analyses across multiple targets. Additionally, we validated the scCHyMErA-Seq results by performing focused qRT-PCR assays for hgRNAs targeting *NRF1* exon-7 and *TAF5* exon-8, demonstrating that exon deletion efficiency correlates with the magnitude of observed phenotypic effects. Finally, we benchmarked scCHyMErA-Seq against Perturb-seq, revealing strong agreement between the two approaches. Collectively, these new data and analyses substantially enhance the depth, rigor, and reliability of our study. Below, we provide a detailed, point-by-point response to each reviewer comment.

Reviewers' Comments:

Reviewer #1:

Remarks to the Author:

Kumari et al. have developed scCHyMErA-Seq to enable exon-resolution functional genomics using hybrid Cas9/Cas12a guide RNAs. The authors extended their recent work employing Cas9/Cas12a hybrids for pooled growth screens to perform scRNA-seq read-outs. After some clever tweaks, they successfully engineered the hybrid guide RNA to contain a capture sequence that appears to work well with Cas12a. They applied their technique to examine the effects of particular exons on the transcriptome, and performed particular validation of one alternative exon, exon 7 in the TF NRF1. In general, the paper reads very well, and the technique seems useful for examining the effects of alternative exons on high resolution phenotypes using scRNA-seq. I have a few major comments about the utility of the technology and some more minor comments about the paper/biology of NRF1.

We thank the reviewer for their positive assessment of our manuscript and for recognizing both the clarity of the presentation and the potential of our approach to study alternative splicing at high resolution.

Major comments:

1: Given that this is a novel screening method, much more discussion is needed re: design of hybrid guide RNAs, and the effect that that guide RNA choice has on guide RNA efficacy. For example: what tools were used to pick and score guide RNAs? How were Cas9 vs. Cas12 guide RNAs chosen (given that you can have 1 of the pair be the 5' guide, and 1 be the 3' guide)? If one were to implement this technique in their own lab, what considerations should be made in guide design? Moreover, in the analyses of their own data, how often do different guide RNA constructs for the same exon agree? What is the variability? How many guide RNA pairs are needed for adequate data interpretation and power? Etcetera. I am not sure that the technology can be directly employed by another group as of yet, which of course should be a major goal when engineering new approaches. The Methods section for the screen itself (experiments) looks ok, but a detailed step-by-step protocol at some point would also be useful to the community.

We agree with the reviewer that our original manuscript did not sufficiently describe strategies for guide RNA selection in scCHyMERa-Seq. This was primarily because we had previously published detailed descriptions of machine-learning approaches for guide scoring, exon deletion library design, and step-by-step protocols for cloning and implementing high-throughput exon deletion proliferation screens using the CHyMERa platform¹⁻³. In the revised manuscript, we now clarify that guide RNAs for the present study were largely selected based on their performance in our previously published fitness screens. We also provide a link to a recently developed web interface (<https://ccr2.cancer.gov/Chymera/>), dubbed CHyMERa-Design, that enables straightforward selection of paired guide RNAs for the deletion of any exon or gene segment of interest. Together with our prior experimental and analytical protocols^{3, 4}, and the detailed documentation of methods, reagents (Addgene link), and analyses provided here (GitHub link), we believe these resources will allow other groups to readily adopt the approach.

The reviewer also raises an important point regarding reproducibility and the variability between different guide RNAs targeting the same exon. To address this, we now include correlation analyses across all guide RNA pairs targeting either genes (for knockout) or exons (for deletion) (**Figures R1A-B**; incorporated as Figs. S7b-c). As expected, gene knockouts using the dual-targeting approach—previously shown to outperform conventional Cas9 single-targeting^{1, 2}—result in very efficient disruption and high concordance between guides (**Figure R1A**). We also observe strong, though somewhat lower, concordance between different guide pairs targeting the same exon (**Figure R1B**). These data demonstrate that independent guide RNAs yield consistent results, supporting the robustness of the method.

R1

A. S7b

B. S7c

Figure R1: Heatmaps illustrating replicate correlations for gene knockout (A) and exon deletion hgRNAs (B). Color intensity reflects Pearson correlation values for differentially expressed genes between the different hgRNA sequences.

With respect to the number of guides, in this study we employed three independent guide pairs per exon. This approach provides sufficient replicates for pseudo-bulk analyses, with each guide serving as a biological replicate, while preserving statistical power for downstream single-cell analyses. Although in principle fewer guides could be used, we do not recommend this, as exon deletions are generally less efficient than gene knockouts. Based on our experience, three independent guide pairs represent a practical balance between cost and data quality. While this is more than the single guide used per gene in genome-scale Perturb-seq studies⁵, we believe the additional investment is warranted to ensure reliable exon-level perturbations.

2: It's still not exactly clear to me why this approach is better than a standard single guide RNA approach when profiling individual exons. In theory, single guide RNAs can still induce mutations in individual exons, effectively resulting in the same information that deleting the exon via Cas9/Cas12 paired guides does. (Clearly if interested in combinatorial targeting of different genes, such an approach is needed, but I remain unconvinced for individual exons). Is this approach much more efficient? Either more discussion of this pre-existing literature and thus rationale, or data from their own screen comparing single guide RNA approaches alone, would be helpful for this point.

We apologize for not clearly explaining the rationale for using a dual Cas9/Cas12a guide approach to delete exons rather than relying on single Cas9 guides that introduce indels within exons. Single-guide targeting of exonic sequences generates indels that can lead to two outcomes: (1) in-frame mutations that alter only a few amino acids while leaving the exon largely intact, or (2) frameshift mutations that disrupt not only the targeted exon but also the entire downstream coding sequence. In both cases, it becomes very difficult to isolate and interpret the specific contribution of the targeted exon to protein function. By contrast, our dual-guide system recruits Cas proteins to flanking intronic sites and precisely excises the exon of interest. This allows us to directly assess the role of individual exons without altering the reading frame or introducing unintended changes to adjacent coding sequences.

To further minimize the possibility of undesired coding disruptions, all guides were designed at least 50 nucleotides away from exon boundaries. Importantly, we have previously demonstrated that control hgRNA pairs in which only one guide targets a flanking intronic sequence while the second guide targets an unrelated intergenic region do not mediate exon deletion². Consistent with this, these partial pairs fail to produce measurable phenotypes and are indistinguishable from dual intergenic controls.² To make this distinction clearer, we have now included a schematic comparing exon disruption via single-guide targeting with exon deletion via dual-guide targeting (**Figure R2**; incorporated as Fig. 1a), and we have expanded the Introduction (lines 59-66, excerpted below) to emphasize why systematic functional interrogation of exons requires a precise deletion strategy.

R2

1a

Figure R2: Schematic overview of the CHyMErA system illustrating its application for dual-gene targeting and precise exon deletion.

“In conventional CRISPR-based gene knockout strategies, single-guide targeting of exonic sequences typically generates indels that can lead either to in-frame mutations producing hypomorphic alleles or to frameshifts that disrupt not only the targeted exon but also the entire downstream coding sequence. Consequently, it is often challenging to isolate and interpret the specific contribution of an individual exon to protein function. In contrast, CHyMERa employs Cas9 and Cas12a nucleases directed to flanking intronic regions, enabling precise excision of the exon of interest. This approach allows direct assessment of exon-specific functions without altering the reading frame (when in-frame exons are targeted) or introducing unintended modifications to neighboring coding sequences^{1,2} (Fig. 1a).”

3: Related to the above: the amount of new biology that the authors have uncovered here should be more extensively examined. How often do exon-level effects disagree with gene-level effects, even in their proof-of-principle library?

We thank the reviewer for raising this important point. Overall, and as expected, we observe a strong correlation between gene knockout and exon deletion effects. As shown in **Figure R3** (incorporated as Fig. 5d), the log₂ fold-change values for differentially expressed genes are concordant between gene knockouts and exon deletions ($r = 0.65$).

R3

5d

Figure R3: Scatter plot comparing log₂ fold-change values of differentially expressed genes across all perturbations, between gene knockouts and exon deletions. Pearson's correlation coefficient is shown.

To more systematically address the reviewer's question, we computed correlations of gene expression changes for each exon deletion compared to the knockout of the corresponding gene. Consistent with the aggregated analysis, the majority of exon deletions (124/224) showed positive correlations (>0.4) with their respective gene knockouts, indicating that exon-level perturbations generally recapitulate gene-level phenotypes.

Among the notable exceptions is *SKP2* exon-6. Deletion of this exon alters the regulation of a gene set distinct from that observed with *SKP2* knockout (**Figure R4A-B**). Interestingly, these discordant genes are enriched for cell cycle-related pathways (**Figure R4C**). Given that *SKP2* promotes degradation of cell cycle regulators such as p27 to facilitate G1–S progression, we hypothesized that these differences might reflect altered cell cycle distributions. Supporting this idea, re-analysis restricted to cells in G1 phase substantially reduced the differences between *SKP2* exon-6 deletion and *SKP2* knockout, and the enrichment for cell cycle pathways was no longer observed (**Figure R4D-G**).

R4

A. 5e

B. 5f

C. 5g

D. S18b

E. S18c

F. S18d

G. S18e

Figure R4: (A, E) Volcano plots of DESeq2 pseudobulk analyses comparing *SKP2* exon-6 deletion with *SKP2* knockout in all cells (A) or in G1-phase cells only (E). Significantly upregulated genes are shown in red and downregulated genes in blue (adjusted $p < 0.05$, $|\log_2 \text{fold-change}| > 0.5$).

(B, F) Scatter plots comparing \log_2 fold-change values of genes between *SKP2* knockout vs. intergenic controls (x-axis) and *SKP2* exon-6 deletion vs. intergenic controls (y-axis). Only genes differentially expressed between *SKP2* knockout and exon-6 deletion are shown. Analyses were performed in all cells (B) or G1-phase cells (F).

(C, G) KEGG pathway enrichment analysis of differentially expressed genes comparing *SKP2* knockout with *SKP2* exon-6 deletion in all cells (C) or G1-phase cells (G). All expressed genes were used as the background set.

(D) Bar plot showing the distribution of cells across different cell cycle phases following transduction with intergenic control hgRNAs, or hgRNAs targeting *SKP2* for knockout or exon-6 deletion. Bars represent mean \pm SD. **** $p < 0.0001$, *** $p < 0.001$, ** $p < 0.01$; one-way ANOVA with Tukey's multiple comparisons test.

Together, these findings indicate that while exon- and gene-level perturbations are generally concordant, individual exons can modulate protein activity in ways that diverge from complete gene loss, producing distinct cell state-dependent expression signatures. These new data are presented in Fig. 5 and Supplementary Fig. 18, and the corresponding Results section has been added at lines 435–472 (included below).

“Exon-level perturbations can diverge from gene-level phenotypes

To assess exon-specific phenotypes that may deviate from corresponding gene-level effects, we first examined the correlation of \log_2 fold-change values between differentially expressed genes across all 224

exon deletions and their respective gene knockouts. As expected, we observed a strong positive correlation ($r = 0.65$), indicating overall concordance between exon deletion and gene knockout effects (**Fig. 5d**).

We next performed differential gene expression analyses directly comparing each exon deletion to the knockout of the corresponding gene (**Supplementary Fig. 18a and Table 7**). As anticipated, the largest discrepancies were observed in cases where gene knockouts elicited broad transcriptional changes (compared to intergenic control), whereas deletion of individual exons caused more subtle effects (**Supplementary Fig. 18a**). This is consistent with the notion that not all exons are essential for the overall gene function.

To systematically identify exons whose perturbation generates transcriptomic profiles distinct from gene knockout effects, we computed correlation coefficients between gene expression changes induced by each exon deletion and the corresponding knockout. Consistent with the aggregate analysis, the majority of exon deletions (124 of 224) showed strong positive correlations (>0.4) with their gene knockouts, indicating that exon-level perturbations typically recapitulate gene-level phenotypes (**Supplementary Table 7**). However, we also identified a subset of exons whose deletion led to reduced correlation with the corresponding knockout while still modulating the expression of a substantial number of genes. A notable example is SKP2 exon-6, whose deletion altered the regulation of a gene set distinct from that affected by SKP2 knockout (**Fig. 5e,f**). The differentially expressed genes unique to the exon-6 deletion are enriched for cell cycle-related pathways (**Fig. 5g**).

SKP2 encodes an E3 ubiquitin ligase that promotes degradation of cell cycle regulators such as p27 to facilitate G1–S transition⁶. In our analyses, both SKP2 knockout and exon-6 deletion result in an increased proportion of cells in the G1 phase (**Fig. 5c and Supplementary Fig. 17b**). Notably, the effect is more pronounced following complete gene knockout compared to exon-6 deletion, consistent with partial loss of SKP2 function upon exon removal (**Supplementary Fig. 18b**). Exon-6 overlaps the leucine-rich repeat domain implicated in substrate recognition, and its deletion is predicted to alter substrate binding. We therefore hypothesized that the observed transcriptional differences could reflect altered cell cycle distributions resulting from impaired SKP2 activity in the absence of exon-6. Indeed, when we re-analyzed differential expression specifically within G1-phase cells, the differences between SKP2 exon-6 deletion and SKP2 knockout were reduced (**Supplementary Figs. 18b,c**), and the enrichment for cell cycle-related terms was largely lost (**Supplementary Fig. 18d**). These findings suggest that exon-level perturbations can fine-tune protein activity in ways that only partially overlap with gene knockout effects, leading to distinct, cell state-dependent transcriptional outcomes.”

4: Was only a single transduction (replicate) of the library performed? When developing a new technology, more replicates are needed to ensure robust and reproducible results.

We appreciate the reviewer’s point regarding replicates. In pooled single-cell CRISPR screens, separate transductions of a gRNA library are generally not required for technical replication, as the pooled format combined with single-cell sequencing provides direct measurement of both the perturbation and the corresponding transcriptional phenotype in each individual cell. This design effectively controls for technical noise in a way that bulk screens cannot. While it is correct that our experiment did not include independent transduction replicates, our dataset profiled over 400,000 cells, of which at least 225,000 were used in the final analysis. On average, this

corresponds to more than 200 cells per individual hgRNA and ~600 cells per targeted exon, with each cell serving as an independent biological replicate for downstream analyses. Consistent with this, UMAP projections demonstrate robust clustering of cells according to the targeted exon. In addition, we employed three independent hgRNAs per exon. Each guide therefore serves as an additional layer of biological replication. As shown in **Figures R1** (Supplementary Fig. 7b-c in the revised manuscript), gene expression changes are highly correlated across independent hgRNAs targeting the same exon, further confirming the reproducibility of the signal. We thus argue that our experimental design is in line with current standards in the field, where pooled single-cell perturbation experiments typically rely on a single transduction and use individual cells as replicates. Importantly, by incorporating multiple hgRNAs per exon, our study goes beyond the minimal standard, providing an additional safeguard for reproducibility⁵.

However, to further confirm the reproducibility of our findings, we performed an independent, focused replicate screen using three hgRNAs targeting NRF1 exon-7, along with hgRNAs targeting intergenic regions, followed by single-cell analysis. The results closely mirrored those from the original scCHyMERa-Seq experiment, thereby validating the robustness and reproducibility of our assay across independent repetitions (**Figure R5A-B**, New Fig. 6a-b).

R5

A. 6a

C. S19a

B. 6b

D. S19b

Figure R5: (A,C) Venn diagrams showing the overlap of differentially expressed genes regulated by perturbation of NRF1 exon-7 as determined by our original scCHyMERa-Seq screen in HAP1 cells and either a small-scale replicate screen in HAP1 cells. Statistical significance was determined by Fisher's exact test. (B) Scatterplots showing the Pearson correlation of log₂ fold-changes for differentially expressed genes identified by scCHyMERa-Seq and a replicate screen for NRF1 exon-7. Each point represents a gene; correlation coefficients with corresponding significance values are indicated. (C) Venn diagram showing the overlap of differentially expressed genes identified

following shared gene inactivations in scCHyMERa-Seq and Perturb-seq. Statistical significance was determined by Fisher's exact test. (D) Scatter plot showing the Pearson correlation of log₂ fold-change values for differentially expressed genes identified by scCHyMERa-Seq and Perturb-seq. Each point represents a gene.

To further assess the reproducibility of our dataset, we compared scCHyMERa-Seq knockout data for 160 genes with corresponding results from the Weissman lab's genome-scale Perturb-seq screen⁵. Despite being performed in different cell lines, editing approaches, and time points, we

observed strong concordance, with a high correlation in differentially expressed genes between the two platforms (**Figure R5C-D**, New Supplementary Fig. 19a-b). It is important to note that unlike our study, the Weissman dataset is based on single gRNA constructs mediating transcriptional repression via CRISPRi rather than gene knockouts. Together, these findings provide independent support for the robustness and reliability of our approach.

The additional Results section describing this finding has been added at lines 477–497 and is also included below:

*“To assess reproducibility and benchmark the performance of scCHyMERa-Seq, we performed an independent single-cell exon deletion experiment using a focused library containing nine intergenic hgRNAs and four hgRNAs targeting NRF1 exon-7 for deletion. Data were analyzed as described above by pseudo-aggregating reads from cells harboring the same hgRNA followed by differential gene expression analysis. In total, we profiled 692 cells containing hgRNAs targeting NRF1 exon-7 for deletion and 4,487 cells carrying intergenic hgRNAs. The resulting profiles revealed a strong overlap in differentially expressed genes between the replicate and the original screen ($p = 5.54e-90$, odds ratio = 30.9, Fisher’s exact test), as well as a high correlation of \log_2 fold-change values ($R = 0.96$; **Figs. 6a,b**), confirming the robustness and reproducibility of the approach.*

*We next utilized our dataset to benchmark scCHyMERa-Seq against orthogonal experimental approaches. First, we compared scCHyMERa-Seq expression profiles with genome-scale Perturb-seq data for genes targeted by both methods⁵. The analysis revealed a significant overlap in differentially expressed genes between the two methodologies ($p < 2.2e-16$, odds ratio = 5.28, Fisher’s exact test; **Supplementary Fig. 19a**). Notably, scCHyMERa-Seq detected a higher number of differentially expressed genes, likely reflecting its more efficient and complete gene inactivation via dual-targeting with Cas9 and Cas12a. This was observed despite the shorter experimental duration for scCHyMERa-Seq (four days post-transduction) compared to Perturb-seq (eight days), which allows approximately twice as long for gene silencing to take effect⁵. The strong concordance between the two approaches is further supported by a high correlation in differential gene expression profiles and a strong positive correlation of \log_2 fold-change values between shared downstream targets for each perturbed gene ($R = 0.76$; **Supplementary Figs. 19b,c**).”*

Minor comments:

5: The mental load of trying to figure out which sequences from Fig 1b are being tested in panels 1c-e is too high. The authors should directly mention which abbreviated construct they are testing in each paragraph for clarity (e.g. “To mitigate this, we incorporated the tevopreQ1 riboswitch to generate CS_preQ1 (yellow bars)” in lines 119-120).

We appreciate the reviewer’s comment regarding the complexity of the presented constructs and thank them for the specific suggestion on how to improve clarity. In the revised manuscript, we have implemented this recommendation, as well as related suggestions from Reviewer 2, to make the figures and text easier to follow. Specifically, we now explicitly reference the abbreviated construct names in the corresponding figure legends and main text, allowing readers to directly connect each construct in Fig. 1b with the results shown in Fig. 1c–e. We believe these revisions substantially improve readability of the section.

6: The doublet rate in Fig S5a seems very high. Can the authors comment on whether this was due to the novel method used, issues with the library cloning or virus prep, or overloading of the 10x library due to quantification issues, or something else?

The elevated doublet rate observed in Fig. S5a reflects an inherent trade-off of the high-throughput 10x Genomics platform. To maximize capture efficiency and cost-effectiveness, the system is intentionally operated at high cell loading, which increases the recovery of single cells but also leads to elevated doublet frequencies that can exceed 25%. Importantly, this effect is independent of our scCHyMERa-Seq methodology, library cloning, or viral preparation. In our experiments, we further minimized the possibility of true biological doublets by transducing the library at a very low multiplicity of infection (MOI < 0.1), which ensures that <1% of cells would be expected to carry two independent hgRNAs.

7: The authors should add some simple QC plots before the UMAPs in Fig 2 that show differences between e.g. targeting hgRNAs and non-targeting (or intronic) hgRNAs.

We thank the reviewer for this helpful suggestion. In the revised manuscript, we have added QC plots that separately assess cells transduced with hgRNAs targeting genes for knockout, exons for deletion, intergenic controls, and non-targeting controls. As shown in the new data (**Figure R6A-C**), incorporated as Supplementary Fig. 5j-l, these analyses demonstrate that the different hgRNA categories yield comparable distributions in UMI counts, number of detected transcripts, and percentage of mitochondrial gene expression per cell.

R6

Figure R6: Box plots summarizing key quality control metrics across cells expressing different categories of hgRNAs. Metrics include the number of unique genes detected per cell (A), total unique molecular identifier (UMI) counts per cell (B), and the percentage of mitochondrial gene expression (C).

8: The authors should validate knockdown of TAF5 with their siRNA used in Fig S10, similar to Fig 4a for NRF1. Also, the authors mention in the legend that the siRNA targets “endogenous” TAF5 (this is also evident for the NRF1 Western) - can they clarify how that is possible (relying on optimized codons in the cDNA to make an “endogenous” specific guide?)

In the revised manuscript, we have added a western blot validating knockdown of TAF5 in the HEK293T Flp-In system. In this experiment, endogenous TAF5 was depleted using an optimized siRNA sequence, and cells were complemented with either full-length TAF5 or the TAF5- Δ ex8 isoform (**Figure R7** incorporated as Supplemental Fig. 12b). To enable selective depletion of the

endogenous protein, the complemented constructs were engineered with wobble mutations in the targeted codons, rendering them resistant to the siRNA. We have also clarified this strategy in the figure legend and methods section to ensure the approach is clearly explained.

R7

S12b

Figure R7: Western blot analysis of TAF5 isoform expression in HEK293 Flp-In cell lines stably expressing doxycycline-inducible 3×FLAG-tagged TAF5 isoforms with exon-8 included (FL) or excluded (ΔE7). The ectopically expressed isoforms contain silent mutations introduced at the siRNA target site, rendering them resistant to siNRF1 treatment. Cells were transfected with either control siRNAs (siNT) or siRNAs targeting endogenous TAF5 (siTAF5). Blots were probed with antibodies against TAF5, FLAG, and GAPDH (loading control).

9: What is the background in the hypergeometric test outlined in Fig S11b? Genes that are not differentially expressed?

We thank the reviewer for pointing out the need for clarification. In the hypergeometric test shown in Fig. S11b (now S13a), the background set consisted of all genes expressed in our RNA-Seq experiment. We have now explicitly stated this in the revised figure legend and text.

10: Can the authors clarify whether the base-editing guides were designed to mutate the donor or acceptor splice site of NRF1 exon-7 in the main text?

To disrupt splicing of NRF1 exon 7, we designed three independent gRNAs: two targeting the acceptor site and one targeting the donor site. This information has now been clarified in the revised manuscript (Line 368).

11: From Fig 4a, it looks like NRF1-ΔE7 is expressed at lower levels than NRF1-FL. This could lead to misinterpretation of results (since lower expression of the alternative construct could look like lower DNA affinity when in fact it's just lower TF concentration). Since the authors use the over-expression constructs instead of the base-edited lines to make their conclusions, either quantification of the Western to indicate similar levels of protein and/or a sentence in the text indicating this caveat would be appropriate.

We thank the reviewer for highlighting this discrepancy, which was also noted by other reviewers. First, we would like to clarify that the base-edited NRF1 exon-7 splice site mutant lines were included in our RNA-Seq analyses, and we observed a highly significant overlap in misregulated events between these lines and the isoform-rescue experiments (Supplementary Fig. 13 f-g). That said, we agree with the reviewer that differences in protein expression could potentially confound interpretation of the ChIP results. To address this, we repeated the ChIP-qPCR experiment after normalizing expression levels of the Δex7 isoform to be equal to the full-length isoform. The results were unchanged: exon 7 skipping led to an almost complete loss of NRF1 recruitment to its target promoters. These new data are provided in **Figures R8A-B** and incorporated as Supplementary Figs. 15b-c in the revised manuscript.

R8

A. S15b

B. S15c

Figure R8: (A) Western blot analysis of NRF1 isoform expression in HEK293 Flp-In cell lines stably expressing doxycycline-inducible 3×FLAG-tagged NRF1 isoforms containing exon 7 (FL) or lacking exon 7 (ΔE7), used in the ChIP-qPCR assays. The ectopically expressed isoforms carry silent mutations at the siRNA target site, rendering them resistant to siNRF1 treatment. Cells were transfected with either control siRNAs (siNT) or siRNAs targeting endogenous NRF1 (siNRF1). Blots were probed

with antibodies against NRF1, FLAG, and GAPDH (loading control). (B) ChIP-qPCR analysis of NRF1-FL and NRF1-ΔE7 binding at promoter regions of the indicated target genes in HEK293 Flp-In cells. Data are shown as mean percent over input ± SEM of three biological replicates. Statistical comparisons between control (siNT) and experimental conditions are indicated; **** p < 0.0001 (two-way ANOVA).

12: The authors mention that exon-7 “overlaps” with the NRF1 DBD and then show an AlphaFold structure in Fig S13d that I find difficult to interpret. Could the authors please add an exon diagram of NRF1, complete with annotated protein domains, to precisely indicate the region that is lost in ΔE7? This should probably be in a main figure.

Thanks for this suggestion. In the revised manuscript, we have added a schematic of the NRF1 protein with annotated domains, highlighting the region encoded by exon 7. This diagram is now included as main Fig. 4a (**Figure R9**) to clearly illustrate the portion of the DNA-binding domain that overlaps exon-7.

R9

4a

Figure R9: Schematic representation of the overlap between NRF1 protein domains and individual exons. The NRF1 protein structure (top) highlights low-complexity regions (orange), the DNA-binding domain (turquoise), and the transcriptional activation domain (green). The splicing schematic of the NRF1 gene (middle) depicts the full-length (FL, top) and exon-7–deleted (ΔE7, bottom) mRNA isoforms. AlphaFold structural predictions^{7, 8} for the NRF1 full-length (top right) and Δexon-7 (bottom right) isoforms are shown, with the region encoded by exon 7 highlighted in red.

13: In the TF field, “transcriptional activity” can often mean the ability to activate or repress transcription (independent of a TF’s ability to bind to DNA). Given the clear role in DNA binding here, the authors may want to steer clear of using the phrase “transcriptional activity” e.g. in line 329.

We thank the reviewer for this helpful comment. To avoid ambiguity, we have removed the term “transcriptional activity” from (previous) line 329 and other instances in the manuscript, replacing it with more precise terminology.

Reviewer #2:

Remarks to the Author:

In this work by Kumari et al., authors present a new method scChymera-Seq, for deleting exons and profiling the effects of alternative exons. Overall the data is high quality and well presented. The text is very well written. The strengths of the study include:

- New method for scRNA-seq from Cas12a-type gRNAs
- Discovery of DRv10 variant of Cas12a improves Cas12a activity and stability
- Knockout controls included for comparison
- Detection of hgRNAs in 92% of cells is impressive
- Cell depth (200 per hgRNA and 600 per exon) is appropriate and robust
- SRSF3 was identified as a regulator of NRF1 splicing using an unbiased screen

We are grateful to the reviewer for their very positive evaluation of our work and for recognizing the strengths of the study. Below, we provide detailed responses to each of the points raised.

A few points require addressing or clarification:

1: Please clarify how Fig 1C and other related figures (e.g. Supp Fig 4c) is normalized – typically Cas9 gRNAs are only detected in ~50% of assayed cells using 10X technology, but in this figure it appears to be 100%. Is data normalized to Cas9 gRNA-detected cells? This is important to clarify because users need to understand the true dropout rate of both Cas9 and Cas12a gRNA detection. The overall efficiency is important to report, rather than only the relative efficiency Cas9 and Cas12a capture as appears to be presented. This additional information could be included in a Supplementary Table

We thank the reviewer for raising this important point. In our capture optimization experiments, Cas9 gRNAs were detected in approximately 80% of profiled cells. However, this likely represents an underestimation due to the relatively low sequencing depth in this dataset, as in our large-scale screen only 4–5% of cells lacked detectable Cas9 gRNAs. In the revised manuscript, we have updated the relevant figures (**Figure R10** and Supplementary Fig. 4d) to display capture rates across all profiled cells, rather than limiting the analysis to those with any gRNA detected. We have also added the overall detection efficiencies for both Cas9 and Cas12a gRNAs in Supplementary Table 1 to provide a comprehensive overview of capture rates.

R10 S4d

Figure R10: Bar plots showing the overall proportions of cells in which Cas9 and/or Cas12a guide gRNAs were successfully detected across all profiled cells using 10x Genomics droplet-based single-cell transcriptomics. The same data is shown as in Fig. 1d, but here the data is normalized to the total of all profiled cells, including those where no gRNAs were captured. Data represent mean \pm SEM from three biological replicates. See also Supplementary Table 1.

2: In Supp Fig 1b, lower right hand panel, it seems the “full length hgRNA” legend of the northern blot also show “w/wo CS” as in the blot to the left? Please correct if so.

We thank the reviewer for catching this oversight. The figure legend has now been corrected to accurately describe the lower right-hand panel in Supplementary Fig. 1b.

3: Please update Supp Fig1 to label each individual panel with a letter. This will help in the text to understand which panel of the northern blot supports each individual claim made

In the revised manuscript we have labeled each individual panel of Supplementary Fig. 1 with letters, and the revised Results section text now refers to these labels for clarity.

4: Please clarify in the main text how the knockout hgRNAs were designed (i.e. were neighboring exons targeted? Upstream exons?)

We thank the reviewer for pointing out this omission. In the revised manuscript, we now explicitly clarify the design strategy for the knockout hgRNAs. Specifically, we selected gRNAs targeting different exons, prioritizing those with high on-target scores and minimal predicted off-target activity. The Results section has also been updated accordingly to specify the design process in more depth (lines 189-213 and below).

“To validate scCHyMErA-Seq, we conducted a screen targeting the deletion of 224 alternative cassette exons from 161 genes. These exons were previously identified as influencing cell fitness in our genetic screens². We focused on frame-preserving exons, which are predicted to produce alternative protein variants², and prioritized those located in genes known to regulate transcriptional phenotypes⁵. We constructed a lentiviral library consisting of 1,066 hgRNAs, with each exon targeted by three independent hgRNAs directing Cas9 and Cas12a nucleases to flanking intronic sites of the exons of interest (Fig. 2a). We selected hgRNAs based on our previous large-scale exon deletion screen², prioritizing independent guide pairs with high on-target scores, minimal predicted off-target activity, and expected deletion sizes below 2 kb. Because only three guides were selected per exon, we did not further optimize the relative positioning of the Cas9 and Cas12a target sites. Notably, our recently developed CHyMErA-Design web tool (<https://ccr2.cancer.gov/CHyMErA/>) facilitates the rational selection of guide pairs for exon deletion screens, allowing simultaneous optimization for efficiency, specificity, and deletion size. In addition, the library includes two independent hgRNAs for gene knockouts, with both Cas9 and Cas12a nucleases inducing mutations in the targeted genes (Fig. 2a). This dual-targeting approach, as previously demonstrated, results in enhanced gene inactivation efficiency^{1, 2}. For this purpose, we randomly selected two of the four independent hgRNAs from our recently developed genome-wide gene knockout CHyMErA library⁹. Guides in this library were rigorously filtered to ensure high on-target efficiency, minimal predicted off-target effects, and comprehensive coverage across all transcript isoforms. Cas9 and Cas12a gRNA pairs were then combined with optimized genomic spacing to target distinct regions of each gene. The library also includes 40 intergenic and 40 non-targeting hgRNAs as negative controls (Supplementary Table 2). HAP1 cells expressing Cas9 and Cas12a were transduced with the library at a low multiplicity of infection (<0.1), and four days later, we performed 10x Genomics droplet-based high-throughput 3' single cell RNA-Seq analysis (Fig. 2b). Illumina sequencing enabled the linkage of gene expression profiles to

their corresponding guide RNA sequences, yielding on average over 47,000 polyadenylated transcript reads and 9,000 CRISPR gRNA reads per profiled cell.”

5: In Fig 4, the effects in the Chip-Seq assay could be caused by reduced levels of the delta Ex7 variant. In the western blot in 4a, the levels appear to be ~50-60% that of the full length isoform. Please confirm whether equal protein expression was observed or not for each isoform, and if unequal, how this was addressed in the assay

To address this concern, we repeated the ChIP assays after confirming that the full-length and Δ exon-7 NRF1 isoforms were expressed at comparable levels (**Figure R8A** and new Supplementary Fig. 15b). Consistent with our initial findings, the Δ exon-7 isoform still showed a markedly reduced ability to be recruited to the promoters of regulated genes, as assessed by qRT-PCR assays (**Figure R8B** and new Supplementary Fig. 15c; see response 11 to reviewer 1).

6: The validation of NRF1 exon skipping using base editing is very nice. However to confirm the utility of the screen, at least one target should be validated at RNA and protein level using the original chimeric screening vector)

Our goal with the NRF1 assays was to demonstrate that scCHyMERa-Seq can capture transcriptional phenotypes consistent with orthogonal exon perturbation approaches, such as splice-site mutations and isoform rescue followed by bulk RNA-Seq. However, we appreciate the reviewer’s suggestion and agree on the importance of validating screen hits using the original chimeric screening vector.

In the revised manuscript, we now include extensive exon deletion validations using scCHyMERa reagents to investigate the role of *NRF1* exon-7 in regulating gene expression, as well as *TAF5* exon-8 and *LSM11* exon-3 in coordinating histone gene expression. Specifically, three independent hgRNAs were cloned for each exon into the scCHyMERa-Seq lentiviral backbone and used to generate exon deletions in HAP1 cells. In all three cases, RT-PCR analyses confirmed robust induction of exon-skipped isoforms. For *TAF5* and *NRF1*, where antibodies detecting both isoforms are available, we also observed increased expression of the skipped isoforms at the protein level (**Figures R11A–B and R12A–B**).

R11

A. 4c

B. S13b

C. 4d

Figure R11: (A) RT-PCR analysis of RNA from HAP1 cells transduced with three independent intergenic control hgRNAs or three independent hgRNAs targeting NRF1 exon-7 for deletion. Constructs compatible with the scCHyMERa-Seq system were used for exon deletion.

(B) Western blot analysis of HAP1 cells transduced with three independent intergenic control hgRNAs or three independent hgRNAs targeting NRF1 exon-7. Exon deletions were generated using constructs

compatible with the scCHyMERa-Seq system. The two NRF1 protein isoforms (full-length, FL; and exon-7-deleted, $\Delta E7$) are indicated. GAPDH served as a loading control.

(C) qRT-PCR validation of scCHyMERa-Seq results following deletion of NRF1 exon-7 using the three independent intergenic control hgRNAs or exon-targeting hgRNAs shown in Fig. 4c. ** $p < 0.01$, *** $p < 0.001$, **** $p < 0.0001$; two-way ANOVA with Dunnett's multiple comparisons test.

For NRF1, we further examined the impact of exon 7 deletion on target genes identified by single-cell and bulk RNA-Seq analysis. All three target genes were validated by qRT-PCR (Figure R11C and new Fig. 4d), as described in lines 350–359 (see below).

“To validate these findings and evaluate the ability of scCHyMERa-Seq to capture individual exons with gene regulatory activity, we cloned the three independent library hgRNAs targeting NRF1 exon-7 for deletion and performed focused assays confirming efficient exon skipping at both the RNA and protein levels (Fig. 4c and Supplementary Fig. 13b). We next examined the effect of NRF1 exon-7 deletion on the expression of three genes identified by scCHyMERa-Seq as regulated by this exon—PUDP, ANKRD26, and YEATS4. Consistent with the single-cell data, deletion of NRF1 exon-7 leads to decreased expression of PUDP and YEATS4 across all three hgRNAs (Fig. 4d). For ANKRD26, only one hgRNA produced a statistically significant effect, although all three showed the same directional trend (Fig. 4d). Notably, the hgRNA yielding the strongest effect was also the one that achieved the highest exon skipping efficiency and the most pronounced expression of the Δ exon-7 protein isoform (Fig. 4c).”

R12

A. 3c

B. 12a

C. 3d

Figure 12: (A) RT-PCR analysis of RNA from HAP1 cells transduced with three independent intergenic hgRNA control sequences or three independent hgRNA sequences targeting *TAF5* exon-8 (top) or *LSM11* exon-3 (bottom) for deletion. Constructs compatible with the scCHyMERa-Seq system were used for exon deletion. (B) Western blot analysis of *TAF5* isoform expression in HEK293 Flp-In cell lines stably expressing doxycycline-inducible 3×FLAG-tagged *TAF5* isoforms with exon-8 included (FL) or excluded ($\Delta E7$). The ectopically expressed isoforms contain silent mutations introduced at the siRNA target site, rendering them resistant to siNRF1 treatment. Cells were transfected with either control siRNAs (siINT) or siRNAs targeting endogenous *TAF5* (siTAF5). Blots were probed with antibodies against *TAF5*, FLAG, and GAPDH (loading control). (C) qRT-PCR analysis of polyadenylated histone gene transcripts following deletion of *TAF5* exon-8 or *LSM11* exon-3 using the three independent intergenic hgRNA controls or exon-targeting hgRNAs shown in Fig. 3c. * $p < 0.05$, ** $p < 0.01$, two-tailed unpaired t-test.

For *TAF5* and *LSM11*, deletion-induced expression of the exon-skipped isoforms was accompanied by increased expression of polyadenylated histone genes, further validating the scCHyMERa-Seq findings (Figure R12C and new Fig. 3d; lines 322–328 and below).

*“To validate these findings, we performed focused assays using the same three hgRNAs from our library to delete *TAF5* exon-8 and *LSM11* exon-3. Efficient skipping of the targeted exons was confirmed for *TAF5* and *LSM11* (Fig. 3c and Supplementary Fig. 12a). Consistent with our scCHyMERa-Seq data, deletion of these exons led to increased expression of histone genes harboring misprocessed transcripts with poly(A) tails (Fig. 3d). These findings were further validated in 293T cells engineered to express either full-length *TAF5* or the *TAF5*- Δ ex8 isoform following siRNA-mediated depletion of endogenous *TAF5* (Supplementary Figs. 12b,c).”*

Finally, we assessed exon deletion efficiency for an additional panel of eight exons using sixteen hgRNAs, providing further support for the robustness of the approach (**Figure R13** and new Fig. 5c). Together with the validations presented in the initial submission, these new data reinforce the suitability of scCHyMERa-Seq for systematic phenotypic profiling of individual exons.

R13
5b

Figure 13: RT-PCR analysis of RNA from HAP1 cells transduced with two independent intergenic control hgRNAs or two independent hgRNAs targeting the indicated exons. Exon deletions were performed using constructs compatible with the scCHyMERa-Seq system.

7: To distinguish the exon-specific deletion phenotypes from complete knockout phenotypes, a more in depth analysis should be performed to identify exon phenotypes that deviate significantly from the knockout phenotypes

We thank the reviewer for this insightful suggestion. This comment is closely related to Reviewer 1's comment 3. We provide the same response below.

Overall, and as expected, we observe a strong correlation between gene knockout and exon deletion effects. As shown in **Figure R3** (see page 4; incorporated as Fig. 5d), the log₂ fold-change values for differentially expressed genes are concordant between gene knockouts and exon deletions ($r = 0.65$).

To more systematically address the reviewer's question, we computed correlations of gene expression changes for each exon deletion compared to knockout of the corresponding gene. Consistent with the aggregated analysis, the majority of exon deletions (124/224) showed positive correlations (>0.4) with their respective gene knockouts, indicating that exon-level perturbations generally recapitulate gene-level phenotypes.

Among the notable exceptions is SKP2 exon-6. Deletion of this exon alters the regulation of a gene set distinct from that observed with SKP2 knockout (**Figure R4A-B**; see page 5). Interestingly, these discordant genes are enriched for cell cycle-related pathways (**Figure R4C**). Given that SKP2 promotes degradation of cell cycle regulators such as p27 to facilitate G1-S progression, we hypothesized that these differences might reflect altered cell cycle distributions. Supporting this idea, re-analysis restricted to cells in G1 phase substantially reduced the differences between SKP2 exon-6 deletion and SKP2 knockout, and the enrichment for cell cycle pathways was no longer observed (**Figure R4D-F**).

Together, these findings indicate that while exon- and gene-level perturbations are generally concordant, individual exons can modulate protein activity in ways that diverge from complete gene loss, producing distinct cell state-dependent expression signatures. These new data are presented in Fig. 5 and Supplementary Fig. 18, and the corresponding Results section has been added at lines 435–472 (included in response to reviewer’s 1 comment 3, pages 5-6).

Reviewer #3:

Remarks to the Author:

CHyMERa-Seq, originally described in Nat Biotech 2020 (PMID: 32249828), detailed in Nat protocols 2021 (PMID: 34508260) and patented in 2020 (US20220348910A1), use a combination of Cas9 and Cas12a/cpf1 to facilitate precise genomic deletion. In this study, Kumari et al made CHyMERa-Seq exon deletion screens they established to be compatible with 10X genomics single-cell capture. Specifically, they optimized capture sequence and the structure of hybrid guide RNA to increase their expression and capture efficiency. Using gene expression as a read out, they demonstrated the transcriptomic changes following 224 alternative cassette (161 genes) exon knockout in cell lines. Furthermore, the authors investigated the function of exon-7 in NRF1. They used ChIP-seq to confirm exon-7 is required for efficient NRF1 recruitment to promoters of its target genes, and pinpoint SRSF3 role on promoting exon-7 inclusion. This is an interesting adoption of combinatorial editing technology they established however demonstrations are not convincing that this adoption could be impactful or widely used in field.

We thank the reviewer for their thoughtful comments and for the opportunity to improve the clarity and impact of our work. However, we respectfully disagree with the assessment that the methodology we present lacks usefulness to the field.

First, we would like to clarify what we see as the relevant field. Our focus is alternative pre-mRNA processing, an area where the functional relevance of the vast majority of splice isoforms remains unknown despite decades of cataloguing efforts. This gap is due largely to the lack of scalable approaches for systematic perturbation of individual exons. The methodology we describe directly addresses this unmet need by enabling high-throughput exon perturbation combined with rich phenotypic readouts offered by single-cell transcriptomics. Contrary to the reviewer's interpretation, scCHyMERa-Seq is not limited to exons of transcription factor genes. In the same way that Perturb-seq has been broadly applied far beyond transcriptional regulators, scCHyMERa-Seq provides a generalizable framework that can be applied to any exon of interest, regardless of gene class.

While we agree that applying this methodology to diverse developmental and disease contexts would be exciting, such applications extend beyond the scope of a methods-focused study. Our goal here is to establish a robust platform that others can adopt for precisely these future investigations. Technically, we highlight that adapting CHyMERa-Seq to the 10x Genomics single-cell platform was not trivial. Achieving simultaneous high editing efficiency and reliable capture of Cas9, Cas12a, and poly(A) transcripts required extensive optimization of gRNA architecture and capture sequences as appreciated by reviewer-2. Our scCHyMERa-Seq method addresses key technical challenges in applying Cas12a to single-cell transcriptomics, establishing a robust foundation for a wide range of CRISPR Cas12a-based screening approaches, including but not limited to transcription factor (genetic) interaction studies.

Finally, as proof of concept, we validated exons with clear biological relevance, including NRF1 exon-7, and additional exons with roles in cell cycle regulation. Together, these examples demonstrate the potential of scCHyMERa-Seq to uncover functionally important exons in a systematic and scalable manner. We therefore believe that scCHyMERa-Seq represents a significant methodological advance for the pre-mRNA processing and functional genomics communities, providing a platform to identify exons with phenotypic impact in a massively parallel manner.

Below, we address the reviewer's specific comments point by point.

1. The basic premise of the manuscript is to define isoform specific impacts due to the loss of an exon. First, the authors should cite recent work to modulate isoform expression using CRISPR by modulating splice site selection (PMID: 38917795). A major concern of the current study is that the authors don't really demonstrate isoform specific impacts of exon loss in regulating gene expression. To determine unique transcriptomic impacts of alternative isoforms, it is crucial to evaluate factors proven to modulate gene expression, such as transcription factors (TFs), especially those with known targets (i.e., ChIP-Seq defined). There are surprisingly few such factors targeted in the current work (30 TFs and chromatin regulators), with only 30 genes that have multiple exons targeted (Supplemental Tables). Only 6 known transcriptional regulators have multiple exons, leaving a small space for evaluating different isoform impacts. However, thousands of TF isoforms have been over-expressed to produce associated scRNA-Seq results (PMC10344468) and ~700 have experimentally defined differential protein-protein (PPI) and protein-DNA interactions (PDI) (PMC12121496, TFisoDB), providing initial ground-truth databases of isoform specific impacts with domain disruption to defining TFs that rewire or negatively regulate transcription. As a result, it is surprising the authors would focus their in-depth validations on a single exon that can easily be predicted to disrupt the DNA binding domain, which should clearly result in a knockout phenotype.

In the revised manuscript we have now cited the work from the Blencowe and Taipale labs (PMID: 38917795), as well as numerous additional studies that have used CRISPR-Cas9 or Cas13-based strategies to modulate isoform expression. We did not include these in the original submission because, unlike our approach, these studies focus on perturbing individual exons in a case-by-case manner rather than establishing a high-throughput and scalable framework for exon perturbation. Nonetheless, we agree they represent important complementary advances, and they are now cited and discussed in the revised manuscript (lines: 536 – 537).

With regard to the reviewer's concern about demonstrating isoform-specific impacts, we respectfully note that the primary goal of this work was to develop and validate a broadly applicable platform—scCHyMErA-Seq—for systematic exon deletion coupled with single-cell transcriptomic readouts. Our focus was not on exhaustively dissecting transcription factor isoforms per se, but on demonstrating the feasibility, efficiency, and scalability of the method. While we agree that large-scale studies dedicated to transcription factor isoforms would be highly valuable, such work represents an independent project requiring substantial time and resources well beyond the scope of the present methods-focused study. The rationale for choosing the exons in this study is based on our previous work on identification of fitness related exons which showed strong enrichment for transcriptional output².

As proof of concept, we chose to validate exons with clear and interpretable functional consequences, including NRF1 exon-7. Although, as the reviewer points out, disruption of the DNA-binding domain can be predicted to impair NRF1 function, our work demonstrates how this can be systematically captured using our screening approach, and further validates the reproducibility of scCHyMErA-Seq through orthogonal assays (including RNA-Seq, ChIP-Seq, and isoform rescue experiments). In addition, we have now extended our validation set to include other exons, including those with roles in cell cycle regulation, to underscore that the method can identify biologically meaningful exons beyond obvious structural motifs.

Taken together, we believe scCHyMErA-Seq provides a generalizable framework for dissecting isoform function across many gene classes—including transcription factors—and anticipate that

the community will adopt it for precisely the kinds of focused isoform-specific questions the reviewer highlights.

2. The authors do not provide evidence that their single-cell based version of CHyMERa-Seq would resolve cell state specific exon dependencies. For example, in Joung et al., above, when evaluating isoform specific impacts of TFs in iPSCs, the perturbations were evaluated following stem cell differentiations to couple TF isoform modulation with cell state regulation. With only 224 alternative cassette, the proposed analyses should be more accurate (decreased dropout) and reduced cost if done simply in 96 wells plates by bulk RNA-seq. It is unclear, why the authors did not evaluate their system by editing Cas9+Cpf1 mouse or embryonic stem cells with differentiation prior to scRNA-Seq.

We agree with the reviewer that applying scCHyMERa-Seq in stem cells coupled to differentiation into neuronal, muscle, or other lineages would be an exciting future application. However, we respectfully note that such experiments fall outside the scope of the present study, which is focused on developing and validating a broadly applicable single-cell exon perturbation platform.

To address the reviewer’s point regarding the ability of scCHyMERa-Seq to resolve cell state-specific dependencies, we re-analyzed our current dataset with a focus on cell cycle variation using established pipelines. This analysis identified 76 genes and 69 exons whose knockout or deletion significantly impacted cell cycle distribution (**Figure R14A-B** and Figs. 5a & 17a). Notably, several of these hits correspond to genes previously validated to regulate the cell cycle (e.g., *TRRAP*^{10, 11}, *SKP2*⁶, and *HEATR1*¹², known to promote G1 arrest upon knockout).

R14

A. 5a

B. S17a

Figure R14: Stacked bar plot showing exons (A) or genes (B) identified by scCHyMERa-Seq whose perturbation affects cell cycle distribution. Cell cycle phases with significantly increased fractions are indicated (** $p < 0.01$, *** $p < 0.001$, **** $p < 0.0001$; Fisher’s exact test, BH-corrected).

To experimentally validate these findings, we assessed five candidate exons predicted to influence the cell cycle and five exons with no apparent effect, using two independent hgRNAs

per exon in HAP1 cells, followed by PI staining and flow cytometry analysis. Across three independent replicates, we confirmed altered cell cycle distribution for 7 out of 10 exons, with the remaining three showing reproducible trends just below statistical significance (**Figure R15** and Fig. 5c). These data underscore that scCHyMERa-Seq is indeed capable of detecting exon perturbations that influence cell state, even within the current proof-of-principle screen. We anticipate that future studies applying this platform in developmentally dynamic systems, such as stem cell differentiation, will provide further powerful insights.

R15

5c

Experimental validation of exon-specific impacts on cell cycle distribution

Figure R15: Validation of scCHyMERa-Seq results by propidium iodide staining and flow cytometry. Stacked bar plots indicate the effect of exon deletions on cell cycle distribution. Cell cycle phases with significant increases are indicated (** $p < 0.01$, * $p < 0.05$; one-way ANOVA).

This new data are described in lines 405–432 of the revised manuscript and are also pasted below:

“scCHyMERa-Seq reveals exons influencing cell cycle distribution

Alternative splicing is a key regulator of cell fate decisions, influencing both cell cycle progression and differentiation^{13–16}. We therefore leveraged the scCHyMERa-Seq dataset to systematically identify exons whose perturbation alters cell cycle phase distributions. Using the Scanpy method for scoring cell cycle marker genes¹⁷, we assigned each single cell in our dataset to a specific cell cycle phase and subsequently aggregated cells based on their genetic perturbation. As expected, cells transduced with intergenic or non-targeting hgRNAs showed comparable distributions, with approximately 17–18% of cells in G1 phase (**Fig. 5a**).

To pinpoint exons that affect cell cycle progression, we grouped cells expressing the same hgRNA and quantified the proportion of cells in each phase. Independent hgRNAs targeting the same exon or gene were treated as biological replicates and compared against intergenic controls. This analysis identified 69 exons and 76 genes whose perturbation significantly altered cell cycle distributions (**Fig. 5a and Supplementary Fig. 17a and Table 6**). Notably, exons within *TRRAP*^{10, 11}, *SKP2*⁶, and *HEATR1*¹² exhibit marked effects—all genes previously implicated in cell cycle regulation.

To validate these findings, we cloned 33 hgRNA constructs from the library, targeting ten exons and the corresponding genes (two-three independent hgRNAs per exon and one per gene), along with two intergenic controls. HAP1 cells were transduced, and exon deletion efficiency was assessed by RT-PCR. All exon-targeting hgRNAs induced efficient exon skipping (**Fig. 5b**). Flow cytometry analysis following propidium iodide staining confirmed that perturbation of seven out of ten tested exons faithfully recapitulated the cell cycle phenotypes identified by scCHyMERa-Seq (**Fig. 5c**). Notably, of the seven exons identified by our focused flow analysis as influencing cell cycle distribution, six are found in genes whose knockdown

similarly affects cell cycle progression according to flow cytometry (**Supplementary Fig. 17b**). Together, these results demonstrate that scCHyMERa-Seq enables systematic identification of exons that influence cell cycle phase distributions, highlighting its potential for linking exon-level perturbations to specific cell states.”

3. The authors do not appear to assess the fidelity of single-cell exon deletion. The exon deletion screen was performed in the HAP1 cell line (near-haploid) which contains only one copy of most chromosomes instead of the usual two. So, in HAP1 cells if gRNA is detected, one could assume the allele has been edited. Any change in exon usage would give a uniform and dominant phenotype. However, this assumption may not apply to other cell types, especially tumor cells where there could be multiple alleles and often not all edited by gRNA. Exon deletion may result varied phenotype that cannot be explained without validating the level of exon exclusion in individual cells. The authors should evaluate using long read sequencing in conjunction with 10X genomics to verify exclusion of target exons, or other appropriate technologies.

We would first like to clarify that the HAP1 cell line used for our screen has undergone diploidization, as confirmed by PI analysis shown below (**Figure R16**). We agree with the reviewer that long-read sequencing in conjunction with 10x Genomics would represent an elegant approach to directly monitor exon deletion at the single-cell level. However, implementing and optimizing such a combined workflow is a substantial undertaking that would require significant time and resources beyond the scope of the present study.

R16

Figure 16: Propidium iodide staining and flow cytometry analysis of cell line ploidy. Shown are 293T, RPE1, and HAP1 cell lines, including the parental clones and the derivatives expressing Cas9 and Cas12a nucleases used in the screen.

That said, we recognize the importance of validating exon skipping efficiency at the RNA level. In line with reviewer #2’s related comment (point 6), we measured exon skipping for 11 representative exons using the same hgRNAs applied in our screen. As shown in the new data (**Figure R13** and Fig. 5b; as well as **Figures R11A & 12A**, pages 15-16), all tested hgRNAs induced substantial exon skipping, albeit with expected differences in efficiency across guides. Importantly, qRT-PCR experiments demonstrated that hgRNAs producing the strongest exon-7 skipping in NRF1 also led to the most pronounced reduction in expression of three validated NRF1 target genes (see **Figure R11** above, page 15). Together, these results provide strong evidence that scCHyMERa-Seq induces exon skipping in a programmable and efficient manner, and that the observed phenotypes are indeed linked to the targeted exon perturbations.

R13**5b**
Figure 13: RT-PCR analysis of RNA from HAP1 cells transduced with two independent intergenic control hgRNAs or two independent hgRNAs targeting the indicated exons. Exon deletions were performed using constructs compatible with the scCHyMERa-Seq system.

4. Throughout the paper, the authors should compare exon loss to not only intergenic non-targeting controls but to the KO of the same gene to determine isoform specific gene expression impacts. A single exon loss can promote or repress PPIs, PDIs or protein-RNA interactions, can only be defined by regulating expression of different isoforms or comparing to knockout. The authors exclude cells that don't have a gRNA, however, including these would further provide evidence that the intergenic targeting control does not induce gene expression changes.

The reviewer raises several important points. Starting with the suggestion to compare intergenic controls with cells lacking hgRNAs, we respectfully disagree that such a comparison would be informative. Cells without detectable gRNAs are unlikely to be truly unperturbed, as they were selected under puromycin pressure and thus should express hgRNAs. Rather, these cells most likely represent cases where gRNA capture failed, leading to a heterogeneous population with unknown perturbations. Instead, we compared cells transduced with intergenic-targeting hgRNAs to those transduced with hgRNAs targeting non-coding regions that are not expected to generate double-strand breaks. These two groups are largely indistinguishable at the gene expression level, supporting the conclusion that intergenic hgRNAs do not confound downstream analyses. We regret that this control analysis was not included in the original submission; it is now presented in **Figure R17** and Supplementary Fig. 7a.

R17**S7a**
Figure 17: Volcano plot showing DESeq2 aggregate differential expression results. Each point represents a gene, comparing cells carrying intergenic control versus non-targeting control hgRNAs.

Regarding the comparison between exon deletions and gene knockouts, we agree that this analysis provides valuable insights into isoform-specific contributions to gene function. We have now performed this analysis (**Figure R18**, Supplementary Fig. 18a and Supplementary Table 7). As expected, the most pronounced differences arise in cases where gene knockouts cause strong and widespread transcriptional changes, whereas deletion of individual exons produces only marginal effects. This outcome is consistent with the fact that not all exons are essential for the overall function of their corresponding genes.

R18
S18a

Figure R18: Bar plot showing the number of differentially expressed genes identified by aggregate pseudobulk analysis across all perturbation types applied in the screen, comparing gene knockout vs exon deletion (green), gene knockout vs intergenic control (blue), and exon deletion vs intergenic control (red).

Notably, we also observed instances where exon deletions produced expression signatures distinct from full gene knockouts. For example, deletion of *SKP2* exon-6 results in altered regulation of a gene set different from that observed with *SKP2* knockout (**Figure R4A-B**). Interestingly, the genes differentially expressed between the gene knockout and exon deletion are enriched for cell cycle–related pathways (**Figure R4C**). Given that *SKP2* promotes degradation of cell cycle regulators such as p27 to facilitate G1–S progression, we hypothesized that these differences could reflect altered cell cycle distributions. Indeed, when we re-analyzed differential expression specifically in G1-phase cells, the differences between *SKP2* exon-6 deletion and *SKP2* knockout were strongly reduced (**Figure R4D-E**), and the enrichment for cell cycle terms was lost (**Figure R4F**). This finding suggests that exon-level perturbations can modulate protein activity in ways that partially overlap, but are not identical to, gene loss—leading to distinct cell state–dependent expression profiles. Together, these analyses reinforce the utility of scCHyMERa-Seq in resolving exon-specific functions and highlight how exon perturbations can produce nuanced and biologically meaningful outcomes that differ from gene-level knockouts. These new data are presented in Fig. 5 and Supplementary Fig. 18, and the corresponding Results section has been added at lines 435–472 (see pages 5-6).

R4

A. 5e

B. 5f

C. 5g

D. S18b

E. S18c

F. S18d

G. S18e

Figure R4: (A, E) Volcano plots of DESeq2 pseudobulk analyses comparing *SKP2* exon-6 deletion with *SKP2* knockout in all cells (A) or in G1-phase cells only (E). Significantly upregulated genes are shown in red and downregulated genes in blue (adjusted $p < 0.05$, $|\log_2 \text{fold-change}| > 0.5$).

(B, F) Scatter plots comparing \log_2 fold-change values of genes between *SKP2* knockout vs. intergenic controls (x-axis) and *SKP2* exon-6 deletion vs. intergenic controls (y-axis). Only genes differentially expressed between *SKP2* knockout and exon-6 deletion are shown. Analyses were performed in all cells (B) or G1-phase cells (F).

(C, G) KEGG pathway enrichment analysis of differentially expressed genes comparing *SKP2* knockout with *SKP2* exon-6 deletion in all cells (C) or G1-phase cells (G). All expressed genes were used as the background set.

(D) Bar plot showing the distribution of cells across different cell cycle phases following transduction with intergenic control hgRNAs, or hgRNAs targeting *SKP2* for knockout or exon-6 deletion. Bars represent mean \pm SD. **** $p < 0.0001$, *** $p < 0.001$, ** $p < 0.01$; one-way ANOVA with Tukey's multiple comparisons test.

5. For all exons, it would be informative to note which domains overlap with the deleted exon and report on any domain that recurrent do or do not show impacts, even though the number of exons targeted is small. Do exons with the most significant impacts relative to KO show large deviations in 3D structure predictions?

In the revised manuscript, we have added a supplementary table (Table S7) indicating whether each targeted exon overlaps with a protein domain, and if so, specifying the domain. Although we do not observe significant enrichment for domain-overlapping exons among those with the

strongest regulatory effects, this may reflect the relatively small number of exons interrogated, as the reviewer notes. A noteworthy example is exon 6 of *SKP2*, discussed above. This exon overlaps the leucine-rich repeat region involved in substrate binding. Its deletion removes 1–2 repeats and is predicted to alter association with specific substrates, although future studies will be required to test this directly.

6. It is not clear if DEGs for exon targeted cells were only computed from the pseudobulks with DESeq2, providing no statistical evaluation of replicates. While there is a strong benefit to aggregating single-cell profiles for each perturbation, this is best done with meta cell approaches to reduce dropouts while retaining replicates for proper statistical evaluation (see Lambourne et al. 2025, PMC12121496).

We apologize for not clearly explaining our pseudobulk analysis in the initial submission. Importantly, our approach does incorporate replicates. Each exon in the library is targeted by three independent hgRNAs. For the pseudobulk analysis, we aggregated reads from cells harboring each hgRNA separately, thereby generating three biological replicates per exon. These were then compared against pseudobulks aggregated from the 40 intergenic control hgRNAs. Thus, our analysis preserves replicate-level information through independent hgRNA sequences while still benefiting from the dropout reduction afforded by pseudobulk aggregation.

7. The GEO entry for the scRNA-Seq does not include a text file with the gRNA and specific exons targeted for each cell barcode.

We thank the reviewer for pointing this out. Although we have included this information in the GitHub page we indeed miss to include it in the GEO entry. We have now included the text file summarizing the cell barcodes corresponding to each hgRNA in the revised manuscript submission (Supplementary Table 9), and will deposit it to GEO as soon as the database's upload function resumes following reopening of the federal government.

Bibliography

1. Gonatopoulos-Pournatzis, T. et al. Genetic interaction mapping and exon-resolution functional genomics with a hybrid Cas9–Cas12a platform. *Nature Biotechnology* **38**, 638-648 (2020).
2. Xiao, M.-S. et al. Genome-scale exon perturbation screens uncover exons critical for cell fitness. *Molecular Cell* **84**, 2553-2572.e2519 (2024).
3. Aregger, M., Xing, K. & Gonatopoulos-Pournatzis, T. Application of CHyMERa Cas9-Cas12a combinatorial genome-editing platform for genetic interaction mapping and gene fragment deletion screening. *Nature Protocols* **2021 16:10 16**, 4722-4765 (2021).
4. Ward, H.N. et al. Analysis of combinatorial CRISPR screens with the Orthrus scoring pipeline. *Nature Protocols* **2021 16:10 16**, 4766-4798 (2021).
5. Replogle, J.M. et al. Mapping information-rich genotype-phenotype landscapes with genome-scale Perturb-seq. *Cell* **185**, 2559-2575.e2528 (2022).
6. Nakayama, K. et al. Targeted disruption of Skp2 results in accumulation of cyclin E and p27(Kip1), polyploidy and centrosome overduplication. *Embo j* **19**, 2069-2081 (2000).
7. Jumper, J. et al. Highly accurate protein structure prediction with AlphaFold. *Nature* **596**, 583-589 (2021).
8. Varadi, M. et al. AlphaFold Protein Structure Database: massively expanding the structural coverage of protein-sequence space with high-accuracy models. *Nucleic Acids Res* **50**, D439-d444 (2022).
9. Behera, A.K. et al. RNA-coupled CRISPR Screens Reveal ZNF207 as a Regulator of LMNA Aberrant Splicing in Progeria. *bioRxiv*, 2025.2004.2025.648738 (2025).
10. Tapias, A. et al. Trrap-dependent histone acetylation specifically regulates cell-cycle gene transcription to control neural progenitor fate decisions. *Cell Stem Cell* **14**, 632-643 (2014).
11. Ichim, G. et al. The histone acetyltransferase component TRRAP is targeted for destruction during the cell cycle. *Oncogene* **33**, 181-192 (2014).
12. Turi, Z., Senkyrikova, M., Mistrik, M., Bartek, J. & Moudry, P. Perturbation of RNA Polymerase I transcription machinery by ablation of HEATR1 triggers the RPL5/RPL11-MDM2-p53 ribosome biogenesis stress checkpoint pathway in human cells. *Cell Cycle* **17**, 92-101 (2018).
13. Zhang, X., Guo, Z., Li, Y. & Xu, Y. Splicing to orchestrate cell fate. *Mol Ther Nucleic Acids* **36**, 102416 (2025).
14. Dominguez, D. et al. An extensive program of periodic alternative splicing linked to cell cycle progression. *eLife* **5** (2016).

15. Gonatopoulos-Pournatzis, T. & Blencowe, B.J. Microexons: at the nexus of nervous system development, behaviour and autism spectrum disorder. *Curr Opin Genet Dev* **65**, 22-33 (2020).
16. Han, H. et al. Systematic exploration of dynamic splicing networks reveals conserved multistage regulators of neurogenesis. *Mol Cell* **82**, 2982-2999.e2914 (2022).
17. Tirosh, I. et al. Dissecting the multicellular ecosystem of metastatic melanoma by single-cell RNA-seq. *Science* **352**, 189-196 (2016).

Response to Reviewers

We thank all reviewers for the time and effort devoted to reviewing our revised manuscript, for their positive assessment, and for their constructive suggestions that have helped strengthen the study.

Reviewer #1 (Remarks to the Author)

I thank the authors for this very thorough response to reviewer comments, which has satisfied all of my previous concerns.

We thank the reviewer for their positive assessment of the revised manuscript.

Reviewer #2 (Remarks to the Author)

The author's responses have addressed all of my concerns. Congratulations on this nice paper.

We thank the reviewer for their positive assessment of the revised manuscript.

Reviewer #3 (Remarks to the Author)

The authors have largely addressed this reviewer's primary concerns. The additional validations (qPCR, western blot), analyses (exon vs. gene, cell cycle) and methodological clarifications (i.e., haploid vs. diploid) substantially improve the rigor and qualitative assessment of the scCHyMERa-Seq.

We thank the reviewer for their positive assessment and for recognizing our efforts to strengthen the manuscript by addressing their comments.

Examples this reviewer was most interested in were those in which exon deletion results in a more significant perturbation than the gene knockout alone, suggesting an isoform with an altered function from the predominantly expressed longer isoform (Figure R18). SKP2 is an intriguing example. Can the authors further comment on ESPL1 exon 8, NUP98 exon 6 and SMG5 exon 5, NUBP1 exon 4 and TFAM exon 5, in the manuscript, which have equivalent or greater perturbation impacts? Specifically, do loss of these exons result in significant impacts to protein expression, 3D protein structure (AlphaFold) or domain composition. Several of these are key mediators of

tumorigenesis in which alternative splicing has not necessarily been implicated in disease pathogenesis. These vignettes will be of most interest to biologists trying to understand, prioritize and model isoform specific impacts in a high throughput manner (these are important examples for others to potentially follow up on in the future).

The reviewer is correct that, in addition to *SKP2* exon-6, several other exons emerge as compelling candidates for isoform-specific functional regulation. We agree that highlighting such examples would be valuable for the research community, particularly for prioritizing alternative splicing events for mechanistic follow-up studies in health and disease.

In response to this suggestion, we have expanded our analysis to include additional exons whose deletion results in stronger transcriptomic perturbations than the corresponding gene knockout. These new analyses are presented in Supplementary Figure 19 (Figure R1 below) and include *SMG5* exon-5, *NUBP1* exon-4, *TFAM* exon-5, and *ESPL1* exon-8. We further discuss these examples in the revised Results section (lines 392-410 also provided below), with emphasis on predicted effects on protein domain architecture and isoform composition. While experimental validation of protein structure or stability for each exon is beyond the scope of this study, we anticipate that these highlighted examples will serve as a useful resource for future in-depth functional investigations.

“Beyond SKP2 exon-6, several additional exons exhibit stronger phenotypic effects upon deletion than the corresponding gene knockouts (Supplementary Fig. 18a), including SMG5 exon-5, NUBP1 exon-4, TFAM exon-5, and ESPL1 exon-8, among others (Supplementary Figs. 19a-h). SMG5 is a core component of the nonsense-mediated decay (NMD) pathway^{1,2} and has been linked to colorectal cancer³. Exon-5 overlaps a tetratricopeptide-like helical domain at the SMG5 N-terminus (Supplementary Figs. 19a, b), that mediates binding to SMG7⁴, suggesting that exon-5 loss may impair SMG7 interaction and impact NMD efficiency. NUBP1 encodes a cytosolic iron-sulfur cluster scaffold with ATPase activity essential for transfer of iron-sulfur clusters to target proteins⁵. Exon-4 overlaps the ATPase domain (Supplementary Figs. 19c, d), raising the possibility that it influences the timing or specificity of protein-protein interactions during iron-sulfur cluster delivery⁶. Finally, TFAM is a mitochondrial transcription factor and high-mobility group (HMG) protein that binds and unwinds mitochondrial DNA and recruits the mitochondrial RNA polymerase to promoters^{7,8}. An exon-5-skipped TFAM isoform was identified previously⁹, and this exon partially overlaps the second HMG domain, suggesting a role in modulating TFAM-DNA binding affinity (Supplementary Figs. 19e, f). Indeed, genes affected by TFAM exon-5 deletion are strongly enriched for mitochondrial pathways related to oxidative phosphorylation. Notably, a previous study identified an ERK-dependent phosphorylation site within exon-5 that reduces TFAM-mediated mitochondrial DNA transcription with implications for Parkinson's disease¹⁰. Together, these vignettes illustrate how scCHyMERa-Seq enables future hypothesis-driven prioritization of splice isoforms for mechanistic studies in health and disease.”

R1 (Supplementary Figure 19)

Figure R1: Exon skipping vignettes with important predicted roles.

(a, c, e, g) Schematics of SMG5 (a), NUBP1 (c), TFAM (e), and ESPL1 (g) illustrating protein domain organization and exon structure. Protein diagrams (top) highlight annotated domains, while splicing

schematics (middle) depict the full-length (FL, top) and exon-deleted (ΔE , bottom) mRNA isoforms. AlphaFold structural predictions^{94, 95} for the full-length (top right) and Δ exon (bottom right) isoforms are shown, with the region encoded by the skipped exon highlighted in red.

(b, d, f, g) Volcano plots showing DESeq2 pseudobulk differential expression analyses comparing cells carrying *SMG5* exon-5 (b), *NUBP1* exon-4 (d), *TFAM* exon-5 (f), or *ESPL1* exon-8 (h), deletions with cells transduced with non-targeting intergenic control hgRNAs. Each point represents a gene; significantly upregulated genes are shown in red and downregulated genes in blue (adjusted $p < 0.05$, $|\log_2$ fold change| > 0.5). p-values were computed using DESeq2's Wald test with Benjamini–Hochberg correction. For each comparison, the most significantly enriched Gene Ontology biological process (GO_BP) term (if any) is indicated separately for upregulated and downregulated genes, considering only terms that contain fewer than 300 genes. Fisher's exact one-tailed test with multiple-testing correction using g:Profiler's g:SCS method.

The reviewer thanks the authors for adding Supplemental Table 7, which details exon deletions that could impact protein domain architecture. In Supplementary Table 1 can the authors include: 1) the longest associated protein coding Ensembl isoform or transcript (where isoform is non-coding) associated with each exon deletion (inferred) and 2) inferred protein length difference from reference isoform (APPRIS principal isoform or equivalent). This is necessary to interpret exon deletion specific or non-specific impacts.

We thank the reviewer for this helpful suggestion, which we agree improves the interpretability of the dataset. As requested, we have updated the Supplementary Tables 7 to include this information. Specifically, for each targeted exon, we now report the primary transcript isoform, identified by intersecting APPRIS principal annotations with MANE Select, as well as the longest associated Ensembl isoform corresponding to the exon-deleted transcript. In addition, we have added a dedicated column indicating the amino acid length difference between the primary isoform and the isoform lacking the targeted exon as well as additional information related to the primary and exon skipped isoforms.

Please include a note in the methods that the HAP1 cells have undergone diploidization.

We thank the reviewer for pointing out this omission. The diploidization of the HAP1 cells is now explicitly stated in the Methods section (lines 1059-1060), and the relevant text is provided below.

“HAP1 cells stably expressing SpCas9 and opCas12a were used for scCHyMErA-Seq experiments¹¹. The HAP1 cell line had undergone diploidization prior to screening.”

References

1. Kishor, A., Fritz, S.E. & Hogg, J.R. Nonsense-mediated mRNA decay: The challenge of telling right from wrong in a complex transcriptome. *Wiley Interdiscip Rev RNA* **10**, e1548 (2019).
2. Kurosaki, T., Popp, M.W. & Maquat, L.E. Quality and quantity control of gene expression by nonsense-mediated mRNA decay. *Nat Rev Mol Cell Biol* **20**, 406-420 (2019).
3. Wang, Y. et al. Nonsense-mediated mRNA decay inhibits TRAF6-dependent anti-tumor immunity in colorectal cancer. *Cell Rep Med* **6**, 102463 (2025).
4. Jonas, S., Weichenrieder, O. & Izaurralde, E. An unusual arrangement of two 14-3-3-like domains in the SMG5-SMG7 heterodimer is required for efficient nonsense-mediated mRNA decay. *Genes Dev* **27**, 211-225 (2013).
5. Netz, D.J., Mascarenhas, J., Stehling, O., Pierik, A.J. & Lill, R. Maturation of cytosolic and nuclear iron-sulfur proteins. *Trends Cell Biol* **24**, 303-312 (2014).
6. Grossman, J.D., Gay, K.A., Camire, E.J., Walden, W.E. & Perlstein, D.L. Coupling Nucleotide Binding and Hydrolysis to Iron-Sulfur Cluster Acquisition and Transfer Revealed through Genetic Dissection of the Nbp35 ATPase Site. *Biochemistry* **58**, 2017-2027 (2019).
7. Barshad, G., Marom, S., Cohen, T. & Mishmar, D. Mitochondrial DNA Transcription and Its Regulation: An Evolutionary Perspective. *Trends Genet* **34**, 682-692 (2018).
8. Kozhukhar, N. & Alexeyev, M.F. 35 Years of TFAM Research: Old Protein, New Puzzles. *Biology (Basel)* **12** (2023).
9. Tominaga, K., Hayashi, J., Kagawa, Y. & Ohta, S. Smaller isoform of human mitochondrial transcription factor 1: its wide distribution and production by alternative splicing. *Biochem Biophys Res Commun* **194**, 544-551 (1993).
10. Wang, K.Z. et al. ERK-mediated phosphorylation of TFAM downregulates mitochondrial transcription: implications for Parkinson's disease. *Mitochondrion* **17**, 132-140 (2014).
11. Xiao, M.-S. et al. Genome-scale exon perturbation screens uncover exons critical for cell fitness. *Molecular Cell* **84**, 2553-2572.e2519 (2024).